# UNIFIED STATIC AND DYNAMIC: TEMPORAL FILTERING NETWORK FOR EFFICIENT VIDEO GROUNDING

## ABSTRACT

Inspired by the activity-silent and persistent activity mechanisms in human visual perception biology, we design a Unified Static and Dynamic Network (UniSDNet), to learn the semantic association between text/audio queries and the video in a cross-modal environment for efficient video grounding. For static modeling, we add the MLP into the residual structure (ResMLP) to handle the global comprehensive interaction between and in the video and multiple queries, achieving mutual semantic supplement. For dynamic modeling, we integrate three characteristics of persistent activity mechanism into network design for a better video context comprehension. Specifically, we construct a diffusive connected video clip graph on the basis of 2D spare temporal masking to reflect the "short-term effect" relationship. We innovatively consider the temporal distance and relevance as the joint "auxiliary evidence clues" and design a multi-kernel **Temporal Gaussian Filter** to expand the joint clue to high-dimensional space, simulating the "complex visual perception", and then conduct element level filtering convolution operations on neighbour clip nodes in message passing stage for finally generating and ranking the candidate proposals. Our UniSDNet is applicable to both *Natural Language Video Grounding(NLVG)* and *Spoken Language Video Grounding(SLVG)* tasks. Our UniSDNet achieves SOTA performance on three widely used datasets for NLVG, as well as datasets for SLVG, *e.g.*, reporting new records at 38.88% $R@1, IoU@0.7$ on ActivityNet Captions and 40.26% $R@1, IoU@0.5$ on TACoS. To facilitate this field, we collect new two datasets (Charades-STA Speech and TACoS Speech) for SLVG. Meanwhile, the inference speed of our UniSDNet is $1.56\times$ faster than the strong multi-query benchmark. We will release the new data and our source code after blind review.

## 1 INTRODUCTION

*Natural Language Video Grounding (NLVG)* Gao et al. (2017); Anne Hendricks et al. (2017) is an important yet challenging task in multimedia understanding, which has drawn increasing attention in recent years due to its vast potential applications in the fields of cross-modal information retrieval Yang et al. (2022) and intelligent Human-Computer Interaction(HCI) Li et al. (2021a). With the development of the Automatic Speech Recognition (ASR) and Text To Speech (TTS), speech is becoming an essential medium for HCI. *Spoken Language Video Grounding (SLVG)* Xia et al. (2022) has also gained a lot of attention. Regardless of whether one employs text or audio queries to locate the video segment, the challenge of video grounding rests in cross-modal semantic understanding.

Existing popular methods solve *NLVG* in two main manners, namely the proposal-based Gao & Xu (2021); Wang et al. (2022); Zheng et al. (2023a) and proposal-free Yuan et al. (2019); Liu et al. (2023a) methods. Proposal-based method utilizes a two-stage strategy of "generate and rank". Proposal-free is a direct regression prediction of the target temporal span based on multimodal features. Regardless of which manner, CNN Xia et al. (2022), RNN Liu et al. (2023a), Transformer Zhang et al. (2021b), GCN Sun et al. (2023) are used for multimodal representation and context learning. For *SLVG* tasks, Xia et al. (2022) adopts the basic framework of proposal-free and uses a video guided audio pre-training model to fuse the semantics of two modalities. Although there has been a lot of achievement on cross-modal semantic understanding, it remains a challenge in the field of video grounding, that is, how to utilize the consistency and complementarity between multimodal information to better understand the semantics of the video in a cross-modal environment.

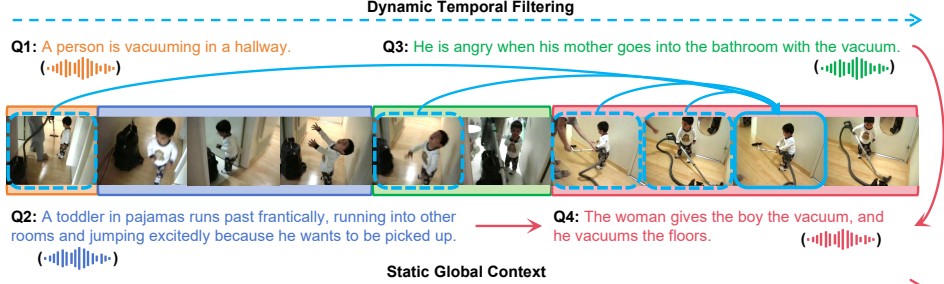

Figure 1: An illustrating example for the video grounding task (query: text or audio). This video is described by four queries (events), all of which have separate semantic context and temporal dependency. Other queries can provide global context (antecedents and consequences) for the current query (*e.g.* query $Q4$). Besides, historical similar scenarios (such as in blue dashed box) help to discover relevant event clues (time and semantic clues) for understanding the current scenario (blue solid box).

We notice a fact that human comes rapidly to understand the queried events in a video, and research shows that in human visual perception biology, this is related to the interplay between activity-silent and persistent activity mechanisms of Prefrontal Cortex (PFC) in the brain Barbosa et al. (2020). Inspired by this, we view the cross-modal interaction of language and video as a combination of static and dynamic interactions. **For static interaction**, we see it as "global broadcast communication" of the brain Volzhenin et al. (2022), which will comprehensively handle multimodal information and learn the semantic connections between them. **For dynamic interaction**, there is a persistent activity mechanism, that is, when viewing videos, human visual perception is transmitted along the Timeline Main Clue and exhibits three characteristics: 1) Short-term Effect; 2) Auxiliary Evidence (Relevance) Clues; 3) Perception Complexity 3.2. These characteristics are extremely important in helping people locate the queried events in the video. Therefore, we integrate the key ideas of these theories into our model design. As shown in Fig. 1, we first comprehensively communicate multiple queries and videos to obtain contextual information for the current query (*e.g.*, $Q4$) and associate different queries to understand video scenes. This process is deemed as a static global interaction. Then we design a visual perception network to imitate dynamic context information transmission in the video. We build a sparely connected relationship (blue arrow in Fig. 1) between video clips to reflect "Short-term Effect", and collect "Evidence (Relevance) Clues" from these neighbour clips (blue dashed box in Fig. 1) by conducting a high-dimensional temporal Gaussian filtering convolution(imitating visual Perception Complexity).

In this paper, we propose a novel **Unified Static and Dynamic Networks (UniSDNet)** (in Fig 3). Specifically, **for the static modeling**, we propose a Static Semantic Supplement Network (S$^3$Net), which contains a purely multi-layer perceptron within residual structure (ResMLP) to capture the association between queries and associate queries with video clips. **For the dynamic modeling**, we design a Dynamic Temporal Filtering Network (DTFNet) based on a Gaussian filtering GCN architecture to capture more useful contextual information in the video sequence. We introduce two important variables 4 for constructing temporal filter between video clip nodes, that is, the relative temporal distance $r$ between nodes and the node relevance weight $a$, and introduce a multi-kernel Temporal Gaussian Filter to extend the joint clue of $r$ and $a$ to high-dimensional space. By performing high-level filtering convolution operations on neighbor nodes, we model fine-grained context correlations between video clips.

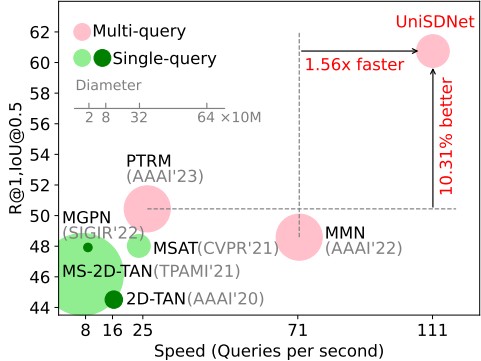

Figure 2: UniSDNet achieves 10.31% of performance gain on the $R@1, IoU@0.5$ metric while being $1.56\times$ faster than multi-query SOTA methods on the ActivityNet Captions dataset. The diameter of the circle indicates the model size (M).

Our main contributions are summarized as follows: (1) We make a new attempt in solving video grounding tasks from the perspective of visual perception biology and propose a Unified Static and Dynamic Networks (UniSDNet), where static module is a fully interactive ResMLP network that provides a global cross-modal environment for multiple queries and the video, a Dynamic Temporal Filter Network (DTFNet) learns the fine context of the video with query attached; (2) We establish an interactive mode for multiple queries and video clips, our method achieves model performance/complexity trade-offs, which has fewer parameters than multi-query methods and achieves a $1.56\times$ faster inference speed than the strong benchmark MMN Wang et al. (2022), which

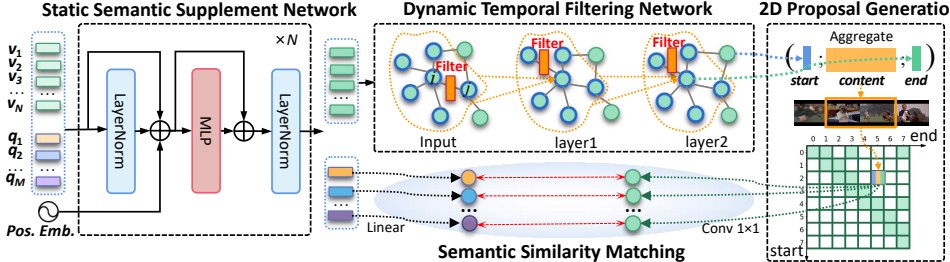

Figure 3: **The architecture of the UniSDNet.** It mainly consists of static and dynamic networks: Static Semantic Supplement Network (S³Net) and Dynamic Temporal Filtering Network (DTFNet). **S³Net** concatenates video clips and multiple queries into a sequence and encodes them through a lightweight single-stream ResMLP network. **DTFNet** is a 2-layer graph network with a dynamic Gaussian filtering convolution mechanism, which is designed to control message passing between nodes by considering temporal distance and semantic relevance as the Gaussian filtering clues when updating node features. The role of 2D temporal map is to retain possible candidate proposals and represent them by aggregating the features of each proposal moment. Finally, we perform semantic matching between the queries and proposals and rank the best ones as the predictions.

shown in Fig. 2; (3) In order to accelerate the research about spoken language video grounding, we collect the new Charades-STA Speech and TACoS Speech datasets with diverse speakers; (4) We conduct experiments on three public datasets for NLVG and three datasets for SLVG, and verify the effectiveness of the proposed method. The SOTA performance on NLVG and SLVG tasks demonstrates the generalization of our model.

## 2    RELATED WORK

**Natural Language Video Grounding (NLVG).** The target of NLVG task is to locate the accurate video moment that semantically corresponds to a specific linguistic query. NLVG is initially pioneered by Gao et al. (2017) and Anne Hendricks et al. (2017), which refers to two main methodological pipelines, *i.e.* proposal-based and proposal-free methods (comparative methods are listed in Sec. 4.2). Herein, 2D-TAN Zhang et al. (2020b) is the first solution depositing possible candidate proposals via a 2D temporal map and MMN Wang et al. (2022) further optimizes it for NLVG. Because of the elegance of 2D-TAN, we incorporate the 2D temporal map into our model, buffering the possible candidate clues. In addition to this, we freshly exploit the relational learning in a video clip graph. Current graph models for NLVG Sun et al. (2023); Gao et al. (2021) overemphasize the correlation between video clip nodes but ignore the intrinsic time-series nature of the video itself. In our work, We fresh sparse masking strategy in 2D-TAN to build a diffusive connected video clip graph with dynamic Temporal Gaussian filtering for video grounding. Extensive experiments in Sec. 4 prove that this artifice is available for both NLVG and SLVG tasks.

**Spoken Language Video Grounding (SLVG).** Xia *et al.* Xia et al. (2022) firstly explore whether the virgin speech rather than language can highlight relevant moments in unconstrained videos and propose the SLVG task. Compared to NLVG, the challenge of SLVG lies in the discretization of speech semantics and the audio-video interaction. The new task demonstrates that text annotations are not necessary to pilot the machine to understand video and proposes a proposal-free curriculum learning method. Recently, with the development of audio pre-training, a breakthrough has been made in the discretization feature representation of speech Baevski et al. (2020; 2022b); Wang et al. (2021). Therefore, in this work we focus on the audio-video interaction challenge of SLVG through the proposed UniSDNet. To facilitate the research of SLVG, we collect two new audio description datasets named Charades-STA Speech and TACoS Speech that originate from Charades-STA Gao et al. (2017) and TACoS Regneri et al. (2013). More details please refer to the Appendix A.1.

## 3    METHODOLOGY

**Task Definition.** Denote the input video as $\mathcal{V} = \{\boldsymbol{v}_1, \boldsymbol{v}_2, \ldots, \boldsymbol{v}_T\} \in \mathbb{R}^{T \times d^v}$, where $d^v$ and $T$ are the feature dimension and total number of video clips. Each video has an annotation set of $\{\mathcal{Q}, \mathcal{M}\}$, in which $\mathcal{Q}$ is a $M$-query set in the text or audio modality and $\mathcal{M}$ represents the corresponding video moments, denoted as $\mathcal{Q} = \{\boldsymbol{q}_1, \boldsymbol{q}_2, \ldots, \boldsymbol{q}_M\} \in \mathbb{R}^{M \times d^q}$ and $\mathcal{M} = \{(t_1^s, t_1^e), (t_2^s, t_2^e), \ldots, (t_M^s, t_M^e)\}$, where $(t_m^s, t_m^e)$ represents the starting and ending timestamps of the $m$-th query, $d^q$ is the dimension of query feature, and $M$ is the query number. The goal of both NLVG and SLVG is to predict the temporal boundary $(t^s, t^e)$ of the queried moment in the video.

### 3.1 STATIC SEMANTIC SUPPLEMENT NETWORK

We first simulate the activity-silent mechanism of human visual perception process, this mechanism manifests as an unconscious "global broadcast communication" of the brain in processing multimodal information Volzhenin et al. (2022). This multimodal information processing optimizes the ability of ephemeral visual perception nerves to capture global information. In multimodal interaction, this global context capture is necessary for representing the video and queries, so we consider the global interaction between the video and queries as a static interaction process. From this perspective, we propose a static semantic supplement network $S^3$Net (in Fig. 3) that adds the MLP into the residual structure (ResMLP). The use of a purely multi-layer perceptron within ResMLP can satisfy this static linear interaction requirement to achieve "global communication". It is an efficient feedforward network with data training and easily reaches model performance/complexity trade-offs. Another advantage of using a linear layer is that it has long-range filters on every layer Touvron et al. (2022).

Firstly, we utilize some pre-trained models to extract original video and query features (C3D Tran et al. (2015), GloVe Pennington et al. (2014), Data2vec Baevski et al. (2022a), *etc.*) and linearly convert them into a unified feature space. Thus, we obtain the video and query features $\boldsymbol{F}_\mathcal{V} \in \mathbb{R}^{T \times d}$ and $\boldsymbol{F}_\mathcal{Q} \in \mathbb{R}^{M \times d}$; $\boldsymbol{F}_{\mathcal{VQ}} = [\boldsymbol{F}_\mathcal{V} || \boldsymbol{F}_\mathcal{Q}] \in \mathbb{R}^{(T+M) \times d}$. Afterwards, we add the position embedding Vaswani et al. (2017) $\boldsymbol{P}_\mathcal{V} \in \mathbb{R}^{T \times d}$ for video feature and $\boldsymbol{P}_\mathcal{Q} \in \mathbb{R}^{M \times d}$ for query feature. Performing an concatenate operation on position embedding as well, there is $\boldsymbol{P}_{\mathcal{VQ}} = [\boldsymbol{P}_\mathcal{V} || \boldsymbol{P}_\mathcal{Q}] \in \mathbb{R}^{(T+M) \times d}$. Let MLPBlock be a combination of a LayerNorm layer, a Linear layer, a ReLU activation layer and a Linear layer. We execute the MLPBlock function to obtain the static interactive video clip features $\hat{\boldsymbol{F}}_\mathcal{V}$ and query features $\hat{\boldsymbol{F}}_\mathcal{Q}$:

$$
\begin{aligned}
\tilde{\boldsymbol{F}}_{\mathcal{VQ}} &= \boldsymbol{F}_{\mathcal{VQ}} + \text{LayerNorm}(\boldsymbol{F}_{\mathcal{VQ}}) + \boldsymbol{P}_{\mathcal{VQ}}, \\
\hat{\boldsymbol{F}}_{\mathcal{VQ}} &= \text{LayerNorm}(\tilde{\boldsymbol{F}}_{\mathcal{VQ}} + \text{MLPBlock}(\tilde{\boldsymbol{F}}_{\mathcal{VQ}})), \\
\hat{\boldsymbol{F}}_\mathcal{V} &= \hat{\boldsymbol{F}}_{\mathcal{VQ}}[1:T; :] \in \mathbb{R}^{T \times d}, \quad \hat{\boldsymbol{F}}_\mathcal{Q} = \hat{\boldsymbol{F}}_{\mathcal{VQ}}[T+1:T+M; :] \in \mathbb{R}^{M \times d}.
\end{aligned}
\tag{1}
$$

Note that UniSDNet leverages multiple queries as the input, facilitating the interaction between the queries themselves, in the video itself (between multiple video clips), and between the queries and video. This allows us to learn the self-modal and cross-modal semantic associations between video and queries without any restrictions, so our model can leverage the complementary effect between multiple queries related to the same video content. We esteem that multiple queries semantics can provide more comprehensive semantic supplementation for the entire video content comprehension.

### 3.2 DYNAMIC TEMPORAL FILTERING GRAPH NETWORK

When browsing videos, besides the static activity-silent mechanism in processing multi-modal information by brain Barbosa et al. (2020), there is also a persistent activity mechanism, that is, human visual perception is transmitted along the **Timeline Main Clue** and exhibits **three characteristics**: **1) Short-term Effect:** nearby perceptions strongly affect current perceptions; **2) Auxiliary Evidence (Relevance) Clues:** semantically relevant scenes in the video provide auxiliary time and semantic clues; **3) Perception Complexity:** the perception process is

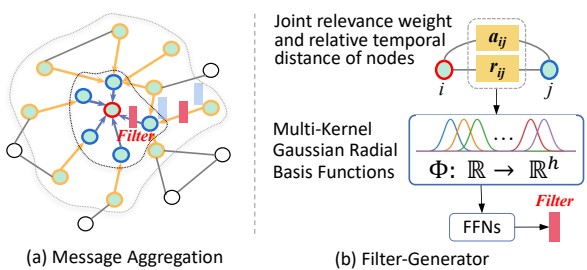

(a) Message Aggregation       (b) Filter-Generator

Figure 4: Dynamic Temporal Filtering Graph.

time-series associative and complex, behaving the characteristics of high-dimensional nonlinearity Barbosa et al. (2020); These characteristics are extremely important in helping people locate the queried events in the video. Graph neural networks have been proven to achieve complex information transmission between nodes Veličković et al. (2017). We simulate people's visual perception process by designing a novel message passing method between video clip nodes and propose a Dynamic Temporal Filtering Graph Network (DTFNet in Figs. 3 and 4).

Specifically, to imitate the Short-term Effect, we construct a diffusive connected graph based on the 2D temporal video clip map (please see "Graph Construction" below). For the discovery of Auxiliary Evidence Clues, we integrate the message passed from each node's neighbours by measuring the

relative temporal distance and the semantic relevance in the graph (please see the filter clue introduced in "How to construct $\mathcal{F}_{filter}$?" below). At last, we use a multi-kernel Gaussian filter-generator to expand the auxiliary evidence clues to a high-dimensional space, simulating human complex visual perception capabilities (please see the filter function in "How to construct $\mathcal{F}_{filter}$?" below).

**Graph Construction.** Let us denote a video graph $\mathcal{G} = (\mathcal{G}_\mathcal{V}, \mathcal{G}_\mathcal{E})$ to represent the relationship in the video $\mathcal{V}$. In graph $\mathcal{G}$, node $\boldsymbol{v}_i$ is the $i$-th video clip and edge $\boldsymbol{e}_{ij} \sim (\boldsymbol{v}_i, \boldsymbol{v}_j) \in \mathcal{G}_\mathcal{E}$ represents whether $\boldsymbol{v}_j$ is $\boldsymbol{v}_i$'s connective neighbor in the graph $\mathcal{G}$. Therein, we utilize a diffusive connecting strategy Zhang et al. (2020b) to construct the adjacency matrix of the graph. In brief, we sparsely connect two clip nodes according to their temporal distance. When the temporal distance of nodes is small, we densely connect them. When the temporal distance becomes long, we skip the edge connection with an interval hyperparameter. With these operations, we obtain a spare adjacency map $\mathcal{G}_\mathcal{E}$. We obatin $\hat{\boldsymbol{F}}_\mathcal{V}$ from the S$^3$Net (in Eq. 1) and take it as the initialization of clip nodes in the graph, namely the initial node embedding of graph is set to $\mathcal{G}_\mathcal{V}^{(0)} = \hat{\boldsymbol{F}}_\mathcal{V} \in \mathbb{R}^{T \times d}$.

**Temporal Filtering Graph Learning.** We build $L$-layer graph filtering convolutions in our implementation. During training, the node embedding $\mathcal{G}_\mathcal{V}^{(l)} = \{\boldsymbol{v}_1^{(l)}, \boldsymbol{v}_2^{(l)}, \cdots, \boldsymbol{v}_T^{(l)}\}$ is optimized at each graph layer, $1 \leq l \leq L$. In this part, we introduce a Gaussian Radial **Filter-Generator** $\mathcal{F}_{filter}$ shown in Fig. 4 to imitate the dynamic flashback process of video for visual perception. There are two core technical difficulties to be resolved below.

**1) How to construct $\mathcal{F}_{filter}$?** Since visual perception is transmitted along the timeline, we consider the relative time between nodes as the primary clue. Besides, similar scenes work appropriately on the comprehension of current scene, so we take into account the semantic relevance between graph nodes as auxiliary clue. Specifically, we compute the two clues of the relative temporal distance $r_{ij}$ of node $v_j$ and node $v_i$ ($r_{ij} = ||j - i||$) and the relevance weight $a_{ij}$ of this two-node pair measured by the $cos(\cdot)$ similarity function, combining them as joint clue $d_{ij} = (1 - a_{ij}) \cdot r_{ij}$. In order to fully reflect the message effect of this joint clue $d_{ij}$ in the model, we expand it to a high-dimensional space (the effect of message transmission for visual perception should actually be infinitely dimensional). Namely, the filter is in the form of $\mathcal{F}_{filter}(d_{ij}) : \mathbb{R} \to \mathbb{R}^h$.

The Gaussian function has been exploited in deep neural networks, such as Gaussian kernel grouping Long et al. (2019), learnable Gaussian fucntion Zheng et al. (2022), Gaussian radial basis function Schütt et al. (2017). Encouragingly, these Gaussian functions have been proven to be effective in simulating the nonlinear information in various scenes. Inspired by this, we design a temporal Gaussian basis function to build the $\mathcal{F}_{filter}$ and expand the joint clue $d_{ij}$ to a high dimension vector $\boldsymbol{f}_{ij} \in \mathbb{R}^h$ in message passing process. We express the form of a single kernel temporal Gaussian as $\Phi(d_{ij}, z) = \exp(-\gamma(d_{ij} - z)^2)$, where $\gamma$ is a hyperparameter and $z$ is a bias we added to avoid a plateau at the beginning of training due to the highly correlated Gaussian filters. Furthermore, we expand it to multiple-kernel Gaussian function to fully represent the complex nonlinear of video perception. Based on the single kernel term, we construct $h$ kernel functions, more studies on the settings of $z$ and $h$ in the Appendix B. The way that we generate the filter $\boldsymbol{f}_{ij}$ of node $\boldsymbol{v}_j$ to node $\boldsymbol{v}_i$ through the multi-kernel Gaussian filer is:

$$\boldsymbol{f}_{ij} = \mathcal{F}_{filter}(d_{ij}) = (\phi_1(d_{ij}), \phi_2(d_{ij}), \cdots, \phi_h(d_{ij})), \quad \phi_i = \Phi(d_{ij}, z_i). \tag{2}$$

**2) How to update the nodes in graph $\mathcal{G}_\mathcal{V}$?** In the stage of message passing on $l$-th layer, we update each node representation by aggregating its neighbor node message to obtain $\mathcal{G}_\mathcal{V}^{(l)}$. For node $v_i$, its neighbor set is $\{\boldsymbol{v}_j \,|\, \boldsymbol{v}_j \in \mathcal{N}(\boldsymbol{v}_i)\}$ corresponding to the adjacency map $\mathcal{G}_\mathcal{E}$. With the multi-kernel Gaussian filter $\boldsymbol{f}_{ij}$, the update of node feature $v_i$ on $l$-th graph layer is described as:

$$v_i^{(l)} = \text{FNN}_2\big( \sum_{\boldsymbol{j} \in \mathcal{N}(\boldsymbol{i})} \text{FNN}_1(v_j^{(l-1)}) \odot \text{FFN}_0(\boldsymbol{f}_{ij}), \tag{3}$$

where $\odot$ represents element-wise multiplication. So far, a video graph with spatiotemporal context correlation of video clips is learned.

### 3.3 Modality Alignment with Semantic Similarity Learning

**2D Proposals Generation.** After obtaining the updated video clip features from the above dynamic graph module, we implement simple moment sampling (Zhang et al., 2020b) on the features to generate a 2D temporary proposal map $\boldsymbol{M}^{2D} \in \mathbb{R}^{T \times T \times d}$ that indicates all candidate moments

(2D Proposals Generation in Fig. 3). The element $\boldsymbol{m}_{ij}$ in the map $\boldsymbol{M}^{\mathrm{2D}}$ indicates the candidate proposal $[\boldsymbol{v}_i, \cdots, \boldsymbol{v}_j]$. For each moment $\boldsymbol{m}_{ij}$, we consider all the clips in the moment interval and the boundary feature is further added to the moment representation as shown in Eq. 4. Afterwards, a stack of 2D convolution is used to encode the moment feature. For more ablation studies about the moment sampling strategy, please refer to the Appendix B.

$$\boldsymbol{m}_{ij} = \mathrm{MaxPool}(\boldsymbol{v}_i^L, \boldsymbol{v}_{i+1}^L, \cdots, \boldsymbol{v}_j^L) + \boldsymbol{v}_i^L + \boldsymbol{v}_j^L \in \mathbb{R}^d, \quad \boldsymbol{M}^{\mathrm{2D}} = CNN(\boldsymbol{m}_{ij}) \in \mathbb{R}^{T \cdot T \cdot d}. \quad (4)$$

**Semantic Similarity Matching.** We calculate the relevance of each query and each moment according to their similarity, generating new 2D moment score maps for the $M$-queries. Specifically, a $1 \times 1$ convolution and a FFN are respectively used to project the moment feature and the query feature (Eq. 1) into the same dimension. Using cosine similarity of the moment and query pair as the measurement of semantic correlation between two modalities, $M$ similarity score maps are computed by

$$\boldsymbol{S}^{\mathcal{M}} = \mathrm{Norm}(\mathrm{Conv2d}_{1 \times 1}(\boldsymbol{M}^{\mathrm{2D}})) \in \mathbb{R}^{T \cdot T \cdot d}, \quad \boldsymbol{S}^{\mathcal{Q}} = \mathrm{Norm}(\mathrm{FNN}(\hat{\boldsymbol{F}}_{\mathcal{Q}})) \in \mathbb{R}^{M \cdot d},$$
$$\tilde{\boldsymbol{S}} = \mathrm{CosSim}(\boldsymbol{S}^{\mathcal{M}}, \boldsymbol{S}^{\mathcal{Q}}) = \{\tilde{s}^1, \tilde{s}^2, \cdots, \tilde{s}^M\} \in \mathbb{R}^{(T \times T) \cdot M}. \quad (5)$$

## 3.4 Training and Inference

Our UniSDNet is a proposal-based method, thus we optimize the score map $\tilde{\boldsymbol{S}}$ with an IoU regression loss and a contrastive learning loss. Following 2D-TAN Zhang et al. (2020b), we first compute the groundtruth IoU Map $\mathrm{IoU}^{\mathrm{GT}} = \{iou^1, iou^2, \cdots, iou^M\} \in \mathbb{R}^{(T \times T) \cdot M}$ corresponding to the queries. That is, we compute the value of intersection over union between each candidate moment and the target moment $(t_{gt}^s, t_{gt}^e)$, and scale this value to (0,1). Hence, the IoU prediction loss is

$$\mathcal{L}_{iou} = \frac{1}{M \cdot T \cdot T} \sum_{m=1}^{M} \sum_{i=1}^{T} \sum_{j=1}^{T} \left( iou_{ij}^m \cdot \log(\tilde{s}_{ij}^m) + (1 - iou_{ij}^m) \cdot \log(1 - \tilde{s}_{ij}^m) \right). \quad (6)$$

We also adopt contrastive learning Wang et al. (2022) as an auxiliary constraint, to fully utilize the positive and negative samples between queries and moments to provide more supervised signals. We define contrastive loss as $\mathcal{L}_{contra}$, details about $\mathcal{L}_{contra}$ are in Appendix A.2. The total loss is $\mathcal{L} = \mathcal{L}_{iou} + \mathcal{L}_{contra}$. The threshold of Non-Maximum Suppression (NMS) is 0.5 during inference.

# 4 Experiments

## 4.1 Datasets and Evaluation Metrics

We conduct extensive experiments on three NLVG benchmarks: ActivityNet Captions Krishna et al. (2017), Charades-STA Gao et al. (2017), and TACoS Regneri et al. (2013) datasets. For SLVG, in addition to the ActivityNet Speech dataset Xia et al. (2022), we have collected two new audio description datasets named Charades-STA Speech and TACoS Speech based on the orignal Charades-STA and TACoS datasets, respectively. More details please refer to Appendix A.1. Following the convention Gao et al. (2017); Mun et al. (2020), we compute the "$R@h, IoU@\mu$" and "$mIoU$" for performance evaluation. Specifically, we set $h \in \{1, 5\}$ and $\mu \in \{0.1, 0.3, 0.5, 0.7\}$. More details please refer to Appendix A.2. We will release the new data and our source code after blind review.

## 4.2 Main Results for Natural Language Video Grounding (NLVG) Task

We compare our UniSDNet with the state-of-the-art methods for **NLVG** and divide them into two groups. **1) Proposal-free methods**: VSLNet Zhang et al. (2020a), LGI Mun et al. (2020), DRN Zeng et al. (2020), CPNet Li et al. (2021b), VSLNet-L Zhang et al. (2021a), BPNet Xiao et al. (2021), VGCL Xia et al. (2022), METML Rodriguez et al. (2023), MA3SRN Liu et al. (2023a). **2) Proposal-based methods**: 2D-TAN Zhang et al. (2020b), CSMGAN Liu et al. (2020), MS-2D-TAN Zhang et al. (2021b), MSAT Zhang et al. (2021b), RaNet Gao et al. (2021), I$^2$N Ning et al. (2021), FVMR Gao & Xu (2021), MMN Wang et al. (2022), MGPN Sun et al. (2022), SPL Liu & Hu (2022), DCLN Zhang et al. (2022), PTRM Zheng et al. (2023a), CRaNet Sun et al. (2023), PLN Zheng et al. (2023b), M$^2$DCapsN Liu et al. (2023b). The best and second-best results are marked in **bold** and underlined.

**Results on the ActivityNet Captions dataset.** The ActivityNet Captions is the largest open domain dataset for NLVG. As shown in Tab. 1, our UniSDNet beats all proposal-free and proposal-based methods at all metrics reported. Specifically, UniSDNet achieves the best 38.88 and 55.47 in terms of

| | Methods | Venue | Text | Video | ActivityNet Captions | | | | | TACoS | | | | |
|---|---|---|---|---|---|---|---|---|---|---|---|---|---|---|
| | | | | | R@1, IoU@ | | R@5, IoU@ | | mIoU | R@1, IoU@ | | R@5, IoU@ | | mIoU |
| | | | | | 0.5 | 0.7 | 0.5 | 0.7 | | 0.3 | 0.5 | IoU0.3 | 0.5 | |
| proposal-free | VSLNet | ACL'20 | GloVe | C3D | 43.22 | 26.16 | - | - | 43.19 | 29.61 | 24.27 | - | - | 24.11 |
| | LGI | CVPR'20 | - | C3D | 41.51 | 23.07 | - | - | 41.13 | - | - | - | - | - |
| | CPNet | AAAI'21 | GloVe | C3D | 40.56 | 21.63 | - | - | 40.65 | 42.61 | 28.29 | - | - | 28.69 |
| | VSLNet-L | TPAMI'21 | Glove | C3D | 43.86 | 27.51 | - | - | 44.06 | 47.11 | 36.34 | - | 36.61 | - |
| | VGCL | ACM MM'22 | GloVe | C3D | 42.96 | 25.68 | - | - | 43.34 | - | - | - | - | - |
| | METML | EACL'23 | BERT | I3D | 43.74 | 27.04 | - | - | 44.05 | - | - | - | - | - |
| | MA3SRN | TMM'23 | GloVe | C3D+Object | 51.97 | 31.39 | 84.05 | 68.11 | - | 47.88 | 37.65 | 66.02 | 54.27 | - |
| proposal-based | 2D-TAN | AAAI'20 | GloVe | C3D | 44.51 | 26.54 | 77.13 | 61.96 | - | 37.29 | 25.32 | 57.81 | 45.04 | - |
| | CSMGAN | ACM MM'20 | GloVe | C3D | 49.11 | 29.15 | 77.43 | 59.63 | - | 33.90 | 27.09 | 53.98 | 41.22 | - |
| | MS-2D-TAN | TPAMI '21 | GloVe | C3D | 46.16 | 29.21 | 78.80 | 60.85 | - | 41.74 | 34.29 | 67.01 | 56.76 | - |
| | MSAT | CVPR'21 | - | C3D | 48.02 | 31.78 | 78.02 | 63.18 | - | 48.79 | 37.57 | 67.63 | 57.91 | - |
| | RaNet | EMNLP'21 | GloVe | C3D | 45.59 | 28.67 | 75.93 | 62.97 | - | 43.34 | 33.54 | 67.33 | 55.09 | - |
| | I$^2$N | TIP'21 | GloVe | C3D | - | - | - | - | - | 31.47 | 29.25 | 52.65 | 46.08 | - |
| | FVMR | ICCV'21 | GloVe | C3D | 45.00 | 26.85 | 77.42 | 61.04 | - | 41.48 | 29.12 | 64.53 | 50.00 | - |
| | MMN | AAAI'22 | DistilBERT | C3D | 48.59 | 29.26 | 79.50 | 64.76 | - | 39.24 | 26.17 | 62.03 | 47.39 | - |
| | MGPN | SIGIR'22 | GloVe | C3D | 47.92 | 30.47 | 78.15 | 63.56 | - | 48.81 | 36.74 | 71.46 | 59.24 | |
| | SPL | ACM MM'22 | GloVe | C3D | 52.89 | 32.04 | 82.65 | 67.21 | - | 42.73 | 32.58 | 64.30 | 50.17 | - |
| | DCLN | ICMR'22 | GloVe | C3D | 44.41 | 24.80 | 74.04 | 56.67 | - | 44.96 | 28.72 | 66.13 | 51.91 | - |
| | CPL | CVPR'22 | GloVe | C3D | 55.73 | 31.37 | 63.05 | 43.13 | - | - | - | - | - | - |
| | PTRM | AAAI'23 | DistilBERT | C3D | 50.44 | 31.18 | - | - | 47.68 | - | - | - | - | - |
| | CRaNet | TCSVT'23 | GloVe | C3D | 47.27 | 30.34 | 78.84 | 63.51 | - | 47.86 | 37.02 | 70.78 | 58.39 | - |
| | PLN | ACM MM'23 | GloVe | C3D | 45.66 | 29.28 | 76.65 | 63.06 | 44.12 | 43.89 | 31.12 | 65.11 | 52.89 | 29.70 |
| | M$^2$DCapsN | TNNLS'23 | GloVe | C3D | 47.03 | 29.99 | 76.64 | 62.83 | - | 46.41 | 32.58 | 66.32 | 52.91 | - |
| | **UniSDNet (Ours)** | | Glove | C3D | 57.67 | 35.64 | 84.46 | 72.47 | 53.68 | 53.59 | 38.34 | 79.01 | 64.83 | 37.54 |
| | **UniSDNet (Ours)** | | DistilBERT | C3D | 60.75 | 38.88 | 85.34 | 74.01 | 55.47 | 55.56 | 40.26 | 77.08 | 64.01 | 38.88 |

Table 1: Comparison with the state-of-the-arts on the *ActivityNet Captions* and *TACoS* datasets for *NLVG*.

| | Methods | Video Feature: VGG | | | | | Video Feature: C3D | | | | | Video Feature: I3D | | | | |
|---|---|---|---|---|---|---|---|---|---|---|---|---|---|---|---|---|
| | | R@1, IoU@ | | R@5, IoU@ | | mIoU | R@1, IoU@ | | R@5, IoU@ | | mIoU | R@1, IoU@ | | R@5, IoU@ | | mIoU |
| | | 0.5 | 0.7 | 0.5 | 0.7 | | 0.5 | 0.7 | 0.5 | 0.7 | | 0.5 | 0.7 | 0.5 | 0.7 | |
| proposal-free | DRN | - | - | - | - | - | 45.40 | 26.40 | 88.01 | 55.38 | - | 53.09 | 31.75 | 89.06 | 60.05 | - |
| | LGI | - | - | - | - | - | - | - | - | - | - | 59.46 | 35.48 | - | - | 51.38 |
| | BPNet | - | - | - | - | - | 38.25 | 20.51 | - | - | 38.03 | 50.75 | 31.64 | - | - | 46.34 |
| | CPNet | - | - | - | - | - | 40.32 | 22.47 | - | - | 37.36 | 60.27 | 38.74 | - | - | 52.00 |
| proposal-based | 2D-TAN | 42.80 | 23.25 | 80.54 | 54.14 | - | - | - | - | - | - | - | - | - | - | - |
| | MS-2D-TAN | 45.65 | 27.20 | 86.72 | 56.42 | - | 41.10 | 23.25 | 81.53 | 48.55 | - | 60.08 | 37.39 | 89.06 | 59.17 | - |
| | FVMR | - | - | - | - | - | 38.16 | 18.22 | 82.18 | 44.96 | - | 55.01 | 33.74 | 89.17 | 57.24 | - |
| | I$^2$N | - | - | - | - | - | - | - | - | - | - | 56.61 | 34.14 | 81.48 | 55.19 | - |
| | CPL | - | - | - | - | - | - | - | - | - | - | 49.05 | 22.61 | 84.71 | 52.37 | - |
| | PLN | 45.43 | 26.26 | 86.32 | 57.02 | 41.28 | - | - | - | - | - | 56.02 | 35.16 | 87.63 | 62.34 | 49.09 |
| | PTRM | 47.77 | 28.01 | - | - | 42.77 | - | - | - | - | - | - | - | - | - | - |
| | CRaNet | 47.12 | 27.39 | 83.51 | 58.33 | - | - | - | - | - | - | 60.94 | 41.32 | 89.97 | 65.19 | - |
| | M$^2$DCapsN | 43.17 | 25.13 | 79.35 | 55.86 | - | 40.81 | 23.98 | 77.93 | 53.52 | - | 55.03 | 31.61 | 84.33 | 63.71 | - |
| | MMN | 47.31 | 27.28 | 83.74 | 58.41 | - | - | - | - | - | - | - | - | - | - | - |
| | MMN† | 47.28 | 27.58 | 83.87 | 57.12 | 42.34 | 42.77 | 24.30 | 82.61 | 53.06 | 39.21 | 53.71 | 31.13 | 85.16 | 59.60 | 46.64 |
| | Improvements | 2.33↑ | 3.85↑ | 1.22↑ | 1.80↑ | 4.89↑ | 15.90↑ | 16.83↑ | 2.53↑ | 10.23↑ | 12.96↑ | 12.21↑ | 25.38↑ | 4.90↑ | 19.43↑ | 13.01↑ |
| | **UniSDNet (Ours)** | 48.41 | 28.33 | 84.76 | 59.46 | 44.41 | 49.57 | 28.39 | 84.70 | 58.49 | 44.29 | 60.27 | 39.03 | 89.33 | 71.18 | 52.71 |

Table 2: Comparison with the state-of-the-arts on the *Charades-STA* dataset for *NLVG*. † denotes the result reproduced by us. Both MMN and our method originate from the exploitation of 2D temporal map. Blue numbers represent the percentage of performance improvement compared with MMN†.

$R@1, IoU@0.7$, and $mIoU$, respectively. In addition to our method, MA3SRN Liu et al. (2023a) has outstanding performance in various metrics, however, it not only leverages both visual features but also object features extracted by Faster-RCNN Ren et al. (2015) for this task. Our UniSDNet merely using C3D features still performs better than MA3SRN, achieving 7.49 and 5.90 improvements in $R@1, IoU@0.7$ and $R@5, IoU@0.7$ metrics, respectively.

**Results on the TACoS dataset.** TACoS (Cooking dataset) has the longest video length (approx. 5 min) and the highest number of events ($>$100) per video (more details in Appendix A.1). As shown in Tab. 1, the proposed UniSDNet achieves the best results across all metrics (*e.g.*, 38.88 on $mIoU$). With the DistilBERT Sanh et al. (2019) as textual feature extraction, our method achives the best results on the $R@1$ metric, *e.g.* 55.56 on $R@1, IoU@0.3$ and 40.26 on $R@1, IoU@0.5$.

**Results on the Charades-STA dataset.** For the Charades-STA dataset, we report the fair comparison results of our method under VGG, C3D and I3D features in Tab. 2. Notably, the duration of the action or event in this datset is relatively short, with an average of 8.09 s, and more subtle human movements need to be identified, resulting in that the models are sensitive to different visual features. Despite under this limitation, for the VGG and C3D visual features, our method achieves the best performance on the stringent metric $R@1$, *e.g.*, 28.33 and 28.39 $R@1, IoU@0.7$ on VGG and C3D feature, respectively. For the I3D video features, our UniSDNet achieves a outstanding record in $R@5, IoU@0.7$, that is, 71.18, demonstrating our the robustness and generalization of our model. Moreover, we specifically make fair comparison of ours with MMN Wang et al. (2022) based on the same 2D temporal proposal map. Compared with MMN, our UniSDNet has the improvements of 4.89↑, 12.96↑, and 13.01↑ in $mIoU$ under VGG, C3D, and I3D features, respectively.

### 4.3 MAIN RESULTS FOR SPOKEN LANGUAGE VIDEO GROUNDING (SLVG) TASK

To our knowledge, the only SLVG work is VGCL Xia et al. (2022), which utilizes CPC Oord et al. (2018) as the audio encoder and C3D as the video encoder to extract audio and visual representations. To facilitate the research of SLVG, we reconstruct the VSLNet and MMN models for the task, and conduct experiments on the new Charades-STA Speech and TACos Speech

| Dataset | Method | Audio Feature | Video Feature | R@1, IoU@ 0.3 | 0.5 | 0.7 | R@5, IoU@ 0.3 | 0.5 | 0.7 | mIoU |
|---------|--------|---------------|---------------|------|------|------|------|------|------|------|
| ActivityNet Speech | VGCL | CPC | | 49.80 | 30.05 | 16.63 | N/A | N/A | N/A | 35.36 |
| | VSLNet | Mel | | 46.75 | 29.08 | 16.24 | N/A | N/A | N/A | 34.01 |
| | VSLNet† | Data2vec | C3D | 51.02 | 30.38 | 17.45 | N/A | N/A | N/A | 37.04 |
| | MMN† | Data2vec | | 51.98 | 35.69 | 20.77 | 85.46 | 75.29 | 56.87 | 37.81 |
| | **UniSDNet** | Data2vec | | **72.27** | **56.29** | **33.29** | **90.41** | **84.28** | **72.42** | **52.22** |
| Charades-STA Speech* | VSLNet† | | I3D | 65.46 | 47.55 | 28.98 | N/A | N/A | N/A | 45.40 |
| | MMN† | Data2vec | VGG | 56.16 | 42.74 | 24.14 | 93.25 | 82.96 | 55.97 | 39.15 |
| | **UniSDNet** | | VGG | 60.73 | 46.37 | 26.72 | 92.66 | 82.31 | 57.66 | 42.28 |
| | **UniSDNet** | | I3D | **67.45** | **53.82** | **34.49** | **94.81** | **87.90** | **69.30** | **48.27** |
| TACoS Speech* | VSLNet† | | | 38.14 | 27.87 | 16.35 | N/A | N/A | N/A | 27.28 |
| | MMN† | Data2vec | C3D | 31.72 | 23.82 | 12.55 | 59.16 | 45.36 | 22.89 | 22.58 |
| | **UniSDNet** | | | **51.66** | **37.77** | **20.44** | **76.38** | **63.48** | **33.64** | **36.86** |

Table 3: Comparison with state-of-the-art methods on the ActivityNet Captions dataset for *SLVG*. ∗ denotes our new collected datasets. † denotes the result reproduced by us. Audio feature 'Mel' refers to the Mel Spectrogram.

datasets. Besides, we consider more features, such as VGG, I3D, and C3D visual features and CPC, Mel, Data2vec audio features. As shown in Tab. 3, it can be seen that our model achieves state-of-the-art performance on all datasets and metrics. Compared with VGCL, our UniSDNet improves 16.66 on $R@1, IoU@0.7$ and 26.24 on $R@1, IoU@0.5$. Compared to the typical proposal-free method VSLNet, our model outperforms it by 2.87 and 9.58 of $mIoU$ on the Charades-STA Speech and TACoS Speech datasets, respectively. For fair comparison with the baseline model MMN, our model still outperforms much better, *e.g.*, 3.13 and 14.28 improvements of $mIoU$ on the Charades-STA Speech and TACoS Speech datasets, respectively.

### 4.4 ABLATION STUDIES

In this section, we conduct in-depth ablation studies to analyse each component of UniSDNet. More discussion about the dynamic module are reported in Appendix B.

**Study on Static and Dynamic Modules.** We remove the static (Sec. 3.1) and dynamic modules (Sec. 3.2) separately to investigate their contribution for cross-modal correlation modeling in our tasks. The results of NLVG and SLVG are reported in Tab. 4. In NLVG, the single static module outperforms the baseline (without static and dynamic modules) by 10.31 and 10.08 improvements in $R@1, IoU@0.7$ and $mIoU$, re-

| Task | Static Module | Dynamic Module | R@1, IoU@ 0.3 | 0.5 | 0.7 | R@5, IoU@ 0.3 | 0.5 | 0.7 | mIoU |
|------|---------------|----------------|------|------|------|------|------|------|------|
| NLVG | ✗ | ✗ | 61.22 | 44.46 | 26.76 | 87.19 | 78.63 | 63.60 | 43.98 |
| | ✓ | ✗ | 73.57 | 58.70 | 37.07 | 91.17 | 85.55 | 73.98 | 54.06 |
| | ✗ | ✓ | 74.56 | 59.45 | 37.44 | 90.98 | 85.43 | 73.60 | 54.43 |
| | ✓ | ✓ | **75.85** | **60.75** | **38.88** | **91.16** | **85.34** | **74.01** | **55.47** |
| SLVG | ✗ | ✗ | 53.63 | 35.91 | 20.51 | 84.71 | 74.21 | 55.95 | 38.23 |
| | ✓ | ✗ | 69.71 | 53.75 | 31.26 | 90.42 | 84.11 | 70.82 | 50.69 |
| | ✗ | ✓ | 71.34 | 54.03 | 31.51 | 89.75 | 82.62 | 68.12 | 50.97 |
| | ✓ | ✓ | **72.27** | **56.29** | **33.29** | **90.41** | **84.28** | **72.42** | **52.22** |

Table 4: Ablation studies on the static and dynamic modules on the ActivityNet Captions and ActivityNet Speech datasets.

spectively. In addition, the single dynamic module exhibits 10.68 and 10.45 improvements than the baseline on $R@1, IoU@0.7$ and $mIoU$, which demonstrates its effectiveness of dynamic temporal modeling in the video. When combining the static and dynamic modules, all the performance metrics are further increased, such as achieving the SOTA records 38.88 in $R@1, IoU@0.7$ and 55.47 in $mIoU$ for NLVG. In SLVG, we observe the similar conclusions. These results demonstrate that both static and dynamic modules indeed have a mutual promoting effect on improving the accuracy.

**Study on Dynamic Filter Graph.** Our dynamic module is a two-layer filtering GCN that differs from existing GCN in the design of message passing. We test five variants of our graph filter, including **1) w/o Graph**, **2) GCN** Kipf & Welling (2016), **3) GAT** Veličković et al. (2017), **4) D**, **5) MLP**. Observing Fig. 5, **w/o Graph** performs the worst. The vanilla **GCN** tracts all the neighbor nodes equally with a convolution operation to aggregate neighbor information. **GAT** is a weighted attention aggregation method. Our method outperforms **GCN** and **GAT** 0.31 and 0.56 on $R@1, IoU@0.7$ for NLVG, and 1.99 and 1.93 on $R@1, IoU@0.7$ for SLVG, respectively. For **D** and **MLP**, we discuss the Gaussian filter setup in our method. In the setting of **D**, we use the the message aggregation wight $f_{ij} = 1/(d_{ij} + 1)$ to replace $\boldsymbol{f}_{ij} = \mathcal{F}_{filter}(d_{ij})$ in Eq. 2, which indicates that we still consider the

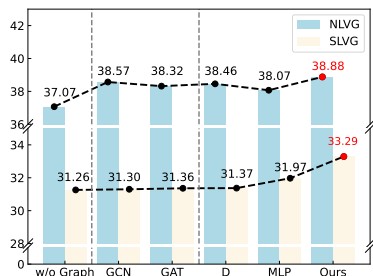

Figure 5: $R@1, IoU@0.7$ results of different message passing strategies in our Graph layer on the ActivityNet Captions and ActivityNet Speech datasets.

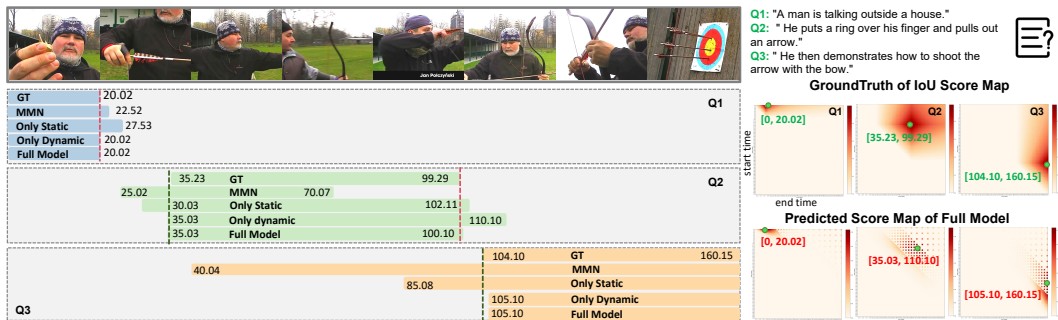

Figure 6: Qualitative examples of our UniSDNet. The right figures display the groundtruth IoU maps and the predicted score maps by our UniSDNet.

same joint clue of temporal distance and relevance between two nodes $d_{ij}$ but remove the entire Gaussian filtering calculation from our method. This replacement results in a decrease of 0.42 and 1.92 on $R@1, IoU@0.7$ for NLVG and SLVG, respectively. **MLP** uses the operation $\text{MLP}(d_{ij})$ to replace the Gaussian basis function $\phi(d_{ij})$ in Eq. 2; in this way, we realize the convolution kernel rather than the Gaussian kernel in the dynamic filter. Compared to **Ours**, **MLP** has a decreased performance of 0.81 and 1.32 on $R@1, IoU@0.7$ for NLVG and SLVG, respectively.

**Results of Model Complexity.** We report the complexity comparison on the ActivityNet Captions dataset. "Infer. Speed" denotes the average inference time per query. Observing Tab. 5, the model size of most single-query methods are obviously lower than that of multi-query methods, whereas multi-query methods exhibit much better results. Returning to the multi-query meth-

| Query | Method | Model Size | Infer. Speed (s/query) | R@1, IoU@0.5 |
|---|---|---|---|---|
| Single | 2D-TAN | 21.62M | 0.061 | 44.51 |
| | MS-2D-TAN | 479.46M | 0.141 | 46.16 |
| | MSAT | 37.19M | 0.042 | 48.02 |
| | MGPN | 5.12M | 0.115 | 47.92 |
| Multi | MMN | 152.22M | 0.014 | 48.59 |
| | PTRM | 152.25M | 0.038 | 50.44 |
| | **UniSDNet (Ours)** | **76.52M** | **0.009** | **60.75** |

Table 5: Complexity comparison on the ActivityNet Captions dataset for NLVG.

ods, they commonly have a large number of model parameters, and our UniSDNet is only half of MMN and PTRM, while our efficiency is improved by 35.71% compared to MMN. Compared to the PTRM method with multiple queries, our accuracy on $R@1, IoU@0.5$ has improved by 10.31.

### 4.5 QUALITATIVE RESULTS

We provide qualitative results of our UniSDNet on the ActivityNet Captions dataset with a video named "v_q81H-V1_gGo" for NLVG, as shown in Fig. 6. MMN Wang et al. (2022) exhibits significant semantic bias, making it impossible to distinguish between $Q2$ and $Q3$. Our **Only Static** accurately predicts the moments, which thanks to the effective static learning of the semantic association between queries and video moments. Our **Only Dynamic** performs well in the three queries too, which thanks to the fine dynamic learning of the video sequence context. The results of the full model **Ours** for all queries are the closest to **Groundtruth (GT)**. It shows that the full model can combine the favorable factors of static (differentiate different query semantics and supplement video semantics) and dynamic (differentiate and associate the related context in the video) modules to obtain more accurate target moment prediction. More challenging and interesting qualitative results for NLVG and SLVG are attached in Appendix C. The quantitative results confirm the effectiveness of our unified static and static methods in solving both NLVG and SLVG tasks.

### 5 CONCLUSION

In this paper, we propose a novel Unified Static and Dynamic Network (UniSDNet) for efficient video grounding. We adopt multi-query mode and achieve model performance/complexity trade-offs, it benefits from both "static" and "dynamic" association between queries and video semantics in a cross-modal environment. We adopt a ResMLP architecture that comprehensively considers mutual semantic supplement through video-queries interaction (static mode). Afterwards, we utilize a dynamic Temporal Gaussian filter convolution to model nonlinear high-dimensional visual semantic perception (dynamic mode). The static and dynamic manners complement each other, ensuring effective 2D temporary proposal generation. We also contribute two new Charades-STA Speech and TACoS Speech datasets for SLVG task. UniSDNet is evaluated on both NLVG and SLVG; for both of them we achieve new state-of-the-art results. We believe that our work is a new attempt and inspire similar video tasks in the design of neural networks guided by visual perception biology.

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

# A  DATASETS AND IMPLEMENTATION DETAILS

## A.1  THE DETAILS OF DATASETS

**The Datasets for NLVG.** The statistics of the datasets ActivityNet Captions Krishna et al. (2017), Charades-STA Gao et al. (2017), and TACoS Regneri et al. (2013) for NLVG are described in Tab 6.

| Datasets | Domain | # Videos | | | # Sentences | | | Average Length | | | Average Queries per Video | | |
|---|---|---|---|---|---|---|---|---|---|---|---|---|---|
| | | Train | Val | Test | Train | Val | Test | Video | Words | Moment | Train | Val | Test |
| ActivityNet Captions | Open | 10,009 | 4,917 | 4,885 | 37,421 | 17,505 | 17,031 | 117.60s | 14.41 | 37.14s | 3.74 | 3.56 | 3.49 |
| Charades-STA | Indoors | 5,336 | - | 1,334 | 12,408 | - | 3,720 | 30.60s | 7.22 | 8.09s | 2.33 | - | 2.79 |
| TACoS | Cooking | 75 | 27 | 25 | 9,790 | 4,436 | 4001 | 286.59s | 9.42 | 27.88s | 130.53 | 164.30 | 160.04 |

Table 6: Data statistics of three widely used datasets in NLVG.

**ActivityNet Captions** contains 19,209 videos from YouTube's open domain collection, which is initially proposed byKrishna et al. (2017) for video captioning and then applied to video grounding. Following the dataset partitioning Zhang et al. (2020b) , we use val_1 as validation set and val_2 as test set. Specifically, there are 37,417, 17,505, and 17,031 sentence-moment pairs for train, validation, and test, respectively. **Charades-STA** contains 9,848 relatively short indoor videos and it is originally from Charades Sigurdsson et al. (2016) used for action recognition and action localization tasks. The dataset is extended by Gao et al. (2017) with language descriptions for *NLVG* task. It has 12,408 and 3,720 sentence-moment pairs for training and testing, respectively. **TACoS** The dataset contains 127 activities that occur in the kitchen, which is built on the MPII-Compositive dataset Rohrbach et al. (2012). We follow the same split in Zhang et al. (2020b), where the training, validation, and testing contain 10,146, 4,589, and 4,083 sentence-moment pairs, respectively. As shown in Tab. 6, **TACoS** has much longer and much more annotated queries for each video, *e.g.*, averaged 286.59s and 130.53 per video in the training set.

**The Datasets for SLVG.** In SLVG task, there is only **ActivityNet Speech dataset** proposed by Xia *et al.* Xia et al. (2022). This dataset is collected based on the ActivityNet Captions dataset Krishna et al. (2017), consists of 37,417 audio-moment pairs, where audio is obtained by volunteers (28 male and 30 female) reading text fluently in a clean environment.

| Datasets | Domain | # Videos | | | # Audios | | | Average Length | | | Audio Source |
|---|---|---|---|---|---|---|---|---|---|---|---|
| | | Train | Val | Test | Train | Val | Test | Video | Duration | Moment | |
| ActivityNet Speech | Open | 10,009 | 4,917 | 4,885 | 37,421 | 17,505 | 17,031 | 117.60s | 6.22s | 37.14s | 58 Volunteers |
| Charades-STA Speech* | Indoors | 5,336 | - | 1,334 | 12,408 | - | 3,720 | 30.60s | 2.33s | 8.09s | Machine Simulation |
| TACoS Speech* | Cooking | 75 | 27 | 25 | 9790 | 4436 | 4001 | 286.59s | 2.89 | 27.88s | Machine Simulation |

Table 7: Data statistics of the datasets used for SLVG. The Charades-STA Speech and TACoS Speech datasets are proposed by us. Both Charades-STA Speech and TACoS Speech datasets are collected by machine simulation Ao et al. (2021) from a 7,931 speaker pronunciation database (the CMU ARCTIC dataset), which has more diverse pronunciations than the AcitivityNet Speech dataset.

To facilitating the research of SLVG, we collect two datasets: **Charades-STA Speech** and **TACoS Speech** datasets. Similar to the collection of ActivityNet Speech dataset Xia et al. (2022), we collect the audio description of corresponding text query. Different from Xia et al. (2022) which employs 58 volunteers to read the text query, we use machine simulator Ao et al. (2021) to read the text query. Specifically, the machine simulator contains 7,931 speaker embeddings with different English pronunciation characteristics from the CMU ARCTIC database[1]. For each query, we randomly select one speaker to generate the audio. Compared to the ActivityNet Speech dataset, the Charades-STA Speech and TACoS Speech datasets collected by us have more diverse pronunciations.

---

[1]CMU ARCTIC database is available at `http://www.festvox.org/cmu_arctic/`

A.2   IMPLEMENTATION DETAILS

**Feature Extraction.**   For a fair comparison, we utilize the same video features provided by 2D-TAN Zhang et al. (2020b), which includes 500-dim C3D feature Tran et al. (2015) on ActivityNet Captions, 4096-dim VGG feature Simonyan & Zisserman (2014) on Charades-STA and 4096-dim C3D feature on TACoS. Besides, there are currently other popular C3D feature and I3D feature Carreira & Zisserman (2017) available on Charades-STA, so we also use the 4096-dim C3D feature from Zeng et al. (2020) and 1024-dim I3D feature provided by Mun et al. (2020). For TACoS dataset, we use PCA dimensionality reduction method Krishna et al. (2017) to reduce the video feature dimension to 500. Following previous work Wang et al. (2022), we use the GloVe Pennington et al. (2014) and DistilBERT Sanh et al. (2019) to extract textual feature. For audio feature, we use the HuggingFace Wolf et al. (2019) implementation of Data2vec Baevski et al. (2022a) with pre-trained model "facebook/data2vec-audio-base-960h" for SLVG. Specifically, we set the audio sampling rate to 16,000 Hz, and use the python audio standard library 'librosa' to read the original audio and input it into the Data2vec model to obtain the audio sequence embedding. Additionally, we use LayerNorm and AvgPool operations to aggregate the entire audio representation. The feature dimensions of both text and audio are 768.

| Datasets | #Clips | Static Module | Dynamic Module | | 2D Proposal Generation | | | #Parameters | | |
|---|---|---|---|---|---|---|---|---|---|---|
| | | Hidden size | #Layers | Hidden size | #Layers | Kernel size | Hidden size | S³Net 3.1 | DTFNet 3.2 | Proposal Generation 3.3 |
| ActivityNet Captions (C3D) | 64 | 1024 | 2 | 256 | 4 | 9 | 512 | 0.53M | 0.68M | 76.79M |
| Charades-STA (VGG) | 16 | 1024 | 2 | 512 | 3 | 5 | 512 | 1.05M | 2.68M | 20.19M |
| Charades-STA (C3D) | 16 | 1024 | 2 | 512 | 3 | 5 | 512 | 1.05M | 2.68M | 20.19M |
| Charades-STA (I3D) | 64 | 1024 | 2 | 256 | 2 | 17 | 512 | 0.53M | 0.68M | 113.91M |
| TACoS (C3D) | 128 | 1024 | 2 | 256 | 3 | 5 | 512 | 0.53M | 0.68M | 16.65M |

Table 8: The hyperparameter settings of UniSDNet model.

**Hyperparameter Settings.**   Tab. 8 shows the hyperparameter settings of UniSDNet. For data preparation, we evenly sample 64 and 128 video clips for ActivityNet Captions dataset with C3D features, and 16, 16, and 64 video clips for Charades-STA dataset with VGG, C3D, and I3D features, respectively. In the static module, we conduct two ResMLP blocks ($N$=2) and the hidden size of feature is set to 1024. In the dynamic module, DTFNet has two graph layers. Therein, we empirically set hyperparameter $\gamma$ to 10.0, the number of Gaussian kernels $h$ to 50, we generate $h$ biases with equal steps starting from 0 and set the step to 0.1. for the dynamic filter $\mathcal{F}_{filter}$. More details about the settings of convolution layers, kernel size, and hidden size for 2D proposal generation are listed in Tab. 8. The parameters size of S³Net (in 3.1), DTFNet (in 3.2) and proposal generation (in 3.3) are also provided in Tab. 8.

**Training Settings.**   We use the AdamW optimizer Loshchilov & Hutter (2017) to optimize the proposed model. For the ActivityNet Captions and TACoS datasets, the learning rate and batch size are set to $8 \times 10^{-4}$ and 12, respectively. For the Charades-STA dataset, we set the learning rate and batch size to $1 \times 10^{-4}$, and 48, respectively. We train the model with 15 epochs on ActivityNet Captions and Charades-STA datasets and 200 epochs on TACoS. All experiments are conducted with a single GeForce RTX 2080Ti GPU.

**Evaluation Metrics.**   Following the convention Gao et al. (2017); Mun et al. (2020), we compute the "$R@h, IoU@\mu$" and "$mIoU$" for performance evaluation. The metric denotes the percentage of correct samples that have at least one correct answer in the top-$h$ choices, where the criterion of the correct answer is that the moment IoU between the predicted result and the groundtruth is greater than a threshold $\mu$. Specifically, we set $h \in \{1, 5\}$ and $\mu \in \{0.1, 0.3, 0.5, 0.7\}$. Also, we use $mIoU$, the average IoU between the prediction and groundtruth across the test set, as an indicator to compare overall performance.

**Details of Contrastive Learning Loss $\mathcal{L}_{contra}$.**   We use noise contrastive estimation Oord et al. (2018) to estimate two conditional distributions $p(q|m)$ and $p(m|q)$. The former represents the probability that a query $q$ matches the video moment $m$ when giving $m$, and the latter represents the probability that a video moment $m$ matches the query $q$ when giving $q$.

$$\mathcal{L}_{contra} = -\left(\sum_{q \in Q^B} \log p(m_q|q) + \sum_{m \in M^B} \log p(q_m|m)\right), \tag{7}$$

where $Q^B$ and $M^B$ are the sets of queries and moments in a training batch. $m_q \in \{m_q^+, m_q^-\}$, $m_q^+$ is the moment matched to the query $q$ (solo positive sample) and $m_q^-$ denotes the moment unmatched

to $\boldsymbol{q}$ in the whole training batch (multiple negative samples). The definition of $\boldsymbol{q}_m \in \{\boldsymbol{q}_m^+, \boldsymbol{q}_m^-\}$ for the moment $\boldsymbol{m}$ is similar to that of $\boldsymbol{m}_q \in \{\boldsymbol{m}_q^+, \boldsymbol{m}_q^-\}$. The objective of contrastive learning is to guide the representation learning of the video and queries and effectively capture mutual matching information between modalities.

### A.3 OVERALL PREDICTION ANALYSIS OF THE MODEL

**The Temporal Distribution of Target Moments.** Fig. 7 shows the temporal distribution of target moments on the ActivtyNet Captions, Charades-STA, and TACoS datasets. The distribution of target moments varies among the these datasets, and our method has good predictive performance than MMN Wang et al. (2022) on all these datasets, indicating that the model has good robustness.

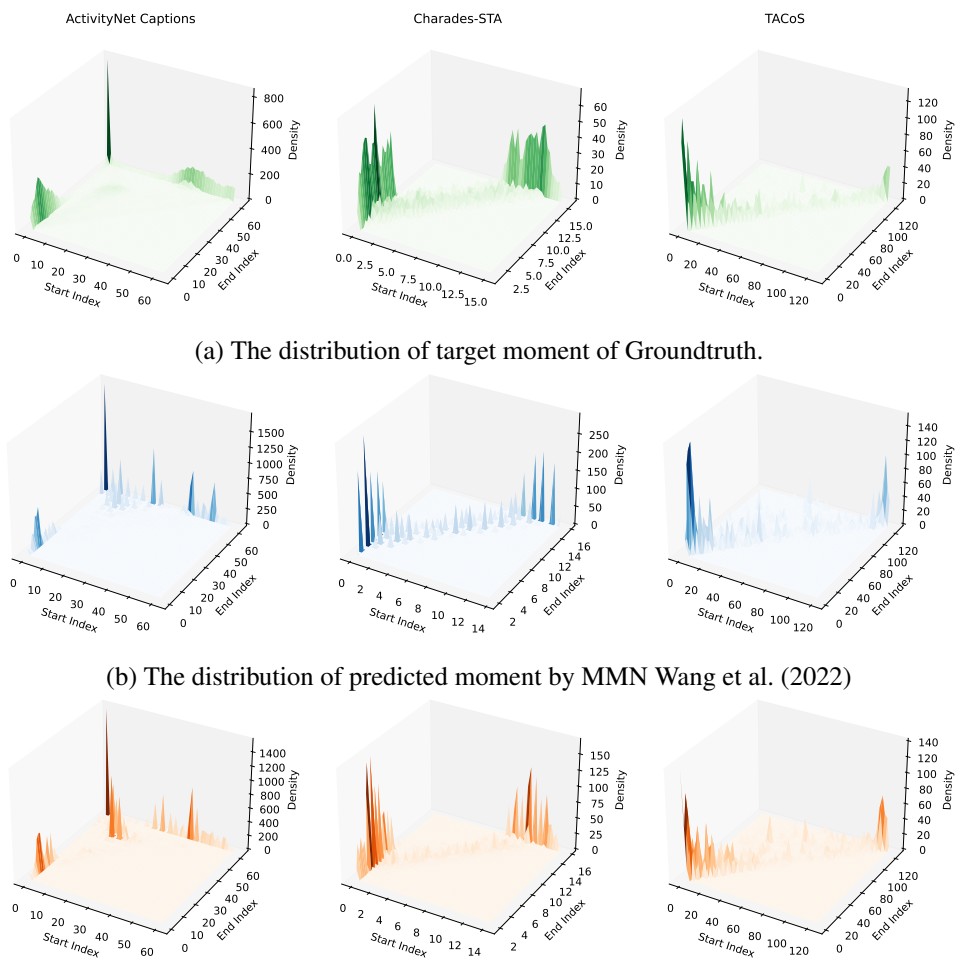

(a) The distribution of target moment of Groundtruth.

(b) The distribution of predicted moment by MMN Wang et al. (2022)

(c) The distribution of predicted moment by UniSDNet

Figure 7: The distribution of predicted moments by our UniSDNet and MMN Wang et al. (2022) on ActivtyNet Captions, Charades-STA, and TACoS datasets.

## B ADDITIONAL ABLATION STUDY RESULTS

**Hyperparameters in the Filter** $\mathcal{F}_{filter}$**.** As introduced in Sec 3.2, the multi-kernel Gaussian basis function involves two important hyperparameters: the number of Gaussian kernels $h$ and the bias parameter $z$. Because we generate $h$ biases with equal steps starting from 0, we study the value of "step" size. Tab. 9 shows the impact of different $h$ and "step" on both NLVG and SLVG. For the task of NLVG, we can find that the model with the same Gaussian kernel number $h$ gives close results even if the Gaussian "step" is different. When the Gaussian kernel number is set to 50 and the Gaussian "step" is set to 0.1, the model achieves the optimal (*i.e.*, $mIoU$ of 55.47). For the task of SLVG, we can get the similar conclusion. We speculate that our method achieves the best results when number of Gaussian kernels $h$ is close to the number of graph nodes.

| Task | #Kernels | Step | R@1, IoU@ | | | R@5, IoU@ | | | mIoU |
|------|----------|------|------|------|------|------|------|------|------|
| | | | 0.3 | 0.5 | 0.7 | 0.3 | 0.5 | 0.7 | |
| NLVG | 25 | 0.1 | 75.12 | 60.20 | 38.02 | 91.20 | 85.82 | 74.68 | 54.91 |
| | 50 | 0.1 | **75.62** | **60.75** | **38.88** | **90.94** | **85.34** | **74.01** | **55.47** |
| | 100 | 0.1 | 74.88 | 59.54 | 38.53 | 91.29 | 85.96 | 74.75 | 54.99 |
| | 200 | 0.1 | 74.28 | 59.60 | 38.62 | 91.33 | 85.91 | 75.05 | 54.96 |
| | 25 | 0.2 | 75.11 | 60.01 | 38.13 | 91.25 | 85.48 | 74.31 | 54.99 |
| | 50 | 0.2 | 75.12 | 60.31 | 38.66 | 90.95 | 85.11 | 73.86 | 55.25 |
| | 100 | 0.2 | 75.30 | 59.73 | 38.47 | 91.65 | 86.07 | 75.16 | 55.13 |
| | 200 | 0.2 | 74.69 | 59.99 | 39.03 | 91.30 | 85.63 | 74.86 | 55.18 |
| SLVG | 25 | 0.1 | 71.35 | 55.71 | 33.70 | 90.44 | 84.42 | 71.53 | 51.91 |
| | 50 | 0.1 | 72.27 | 56.29 | 33.29 | 90.41 | 84.28 | 72.42 | 52.22 |
| | 100 | 0.1 | 71.22 | 55.65 | 33.16 | 90.36 | 84.12 | 71.57 | 51.72 |
| | 200 | 0.1 | **72.64** | **56.76** | 31.27 | 90.32 | **84.29** | 70.80 | 51.65 |
| | 25 | 0.2 | 71.77 | 55.45 | 31.85 | 90.36 | 84.16 | 71.08 | 51.28 |
| | 50 | 0.2 | 71.48 | 55.23 | 33.29 | 90.45 | 84.15 | 71.64 | 51.78 |
| | 100 | 0.2 | 73.31 | 55.88 | 31.28 | 90.23 | 84.03 | 71.38 | 51.85 |
| | 200 | 0.2 | 71.79 | 55.92 | 33.76 | 90.16 | 84.00 | 70.89 | 51.91 |

Table 9: Ablation study results of Gaussian kernel number $h$ and step $z$ on the ActivityNet Captions and ActivityNet Speech datasets.

**Effect of Dynamic Graph Layer.** We investigate the influence of the graph layer of our dynamic module. As shown in Fig. 8, we observe that our model achieves the best result (*e.g.*, $R@1, IoU@0.7$ is 38.88) when the total number of graph layer is set to 2. It is speculated that on the basis of informative context modelling by the static module, two-layers dynamic graph module is enough for relational learning of the video. Additionally, graph convolutional networks generally experience over-smoothing problem as the number of layers increases, leading to a performance decline Li et al. (2018). Our model exhibits good stability on the 1~6-th graph layers.

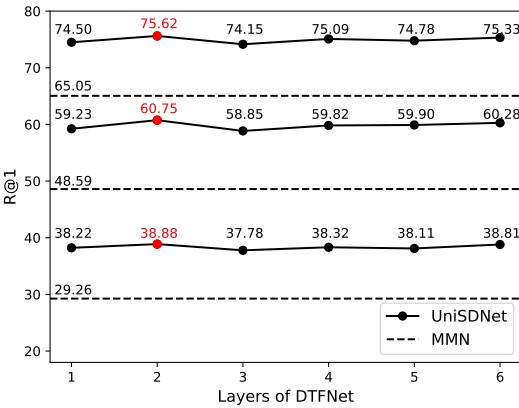

Figure 8: The results across different graph layer on the AcvitivtyNet Captions dataset for NLVG. From top to bottom, the metrics are $R@1, IoU@0.3, IoU@0.5$, and $IoU@0.7$, respectively.

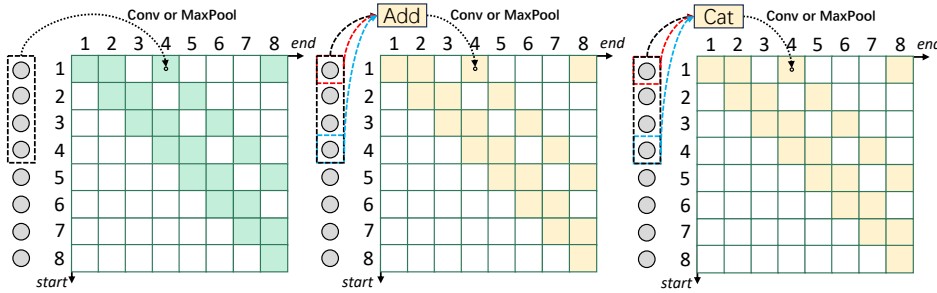

Figure 9: Different feature sampling strategies for 2D proposal generation. (a) Only the features of the proposal (taken as content) are used for proposal generation. (b) The content and boundary features are fused by addition operation. (c) The content and boundary features are fused with concatenation operation.

**Effect of Feature Sampling for 2D Proposals Generation.** To discuss the sensitivity of feature sampling strategy for 2D proposals generation, we evaluate the effects of moment content and boundary features in this section. As shown in Fig. 9, we conduct experiments with different proposal generation strategies: (a) only the features of proposal (taken as content) are used to generate proposal; (b) the content and boundary features are summed together to generate proposal; (c) the content and boundary features are concatenated together to generate proposal. The 2D proposals can be generated by 1D Conv or MaxPool Zhang et al. (2020b). The experiments conducted on both NLVG and SLVG tasks are shown in Tab. 10. For NLVG, we can see that the model with MaxPool strategy achieves better performance than convolution strategy, $e.g.$, 38.99 $vs.$ 38.20 in terms of $R@1, IoU@0.7$ when the model using content feature. In addition, the addition operation performs better than the concatenation operation, $e.g.$, 55.47 $vs.$ 54.98 when the model using the content and boundary features. For SLVG, we can get the similar conclusion as well as NLVG. Therefore, we use both the content and boundary features to generate proposal through MaxPool and Conv for NLVG and SLVG, respectively.

| Task | Generation | Features | Fusion | R@1, IoU@ | | | R@5, IoU@ | | | mIoU |
|------|-----------|----------|--------|------|------|------|------|------|------|------|
| | | | | 0.3 | 0.5 | 0.7 | 0.3 | 0.5 | 0.7 | |
| NLVG | Conv | Content | N/A | 75.30 | 60.27 | 38.20 | 90.86 | 85.16 | 73.17 | 55.13 |
| | Conv | Content, Boundary | Addition | 75.85 | 60.70 | 38.75 | 90.85 | 85.05 | 73.25 | 55.41 |
| | Conv | Content, Boundary | Concatenation | 74.76 | 60.30 | 38.80 | 90.70 | 84.96 | 73.00 | 55.15 |
| | MaxPool | Content | N/A | 75.62 | 60.40 | 38.99 | 90.94 | 85.22 | 73.97 | 55.39 |
| | MaxPool | Content, Boundary | Addition | **75.85** | **60.75** | 38.88 | **91.16** | **85.34** | **74.01** | **55.47** |
| | MaxPool | Content, Boundary | Concatenation | 75.13 | 59.96 | 38.25 | 91.26 | 85.59 | 73.91 | 54.98 |
| SLVG | Conv | Content | N/A | 71.02 | 55.24 | 32.88 | 90.38 | 84.25 | 71.38 | 51.66 |
| | Conv | Content, Boundary | Addition | 72.27 | 56.29 | 33.29 | 90.41 | 84.28 | 72.42 | **52.22** |
| | Conv | Content, Boundary | Concatenation | 71.45 | 55.79 | 33.20 | 90.55 | 84.16 | 71.48 | 51.76 |
| | MaxPool | Content | N/A | 71.26 | 55.25 | 33.74 | 90.49 | 84.29 | 72.46 | 51.80 |
| | MaxPool | Content, Boundary | Addition | **72.60** | **56.64** | 32.61 | **90.82** | **84.89** | **72.48** | 52.04 |
| | MaxPool | Content, Boundary | Concatenation | 69.85 | 53.96 | 32.05 | 90.36 | 84.12 | 72.24 | 50.68 |

Table 10: Comparison of different proposal generation methods.

## C   MORE VISUALIZATION OF PREDICTION RESULTS

In order to clearly demonstrate the specific role of our proposed unified static and dynamic networks in cross-modal video grounding, we provide more challenging visualization cases in this section as a supplement to Sec. 4.5.

### C.1   VISUALIZATION ON ACTIVITYNET CAPTIONS FOR NLVG

**Video Sample with Complex Scene Transitions.** As shown in Fig 10 (a), there are multiple scene transitions in video sample "ID: v_rKtktLDSOpA" from the ActivityNet Captions dataset and different events have serious intersection in the temporal sequence of video. For example, there is an intersection between the end of the moment corresponding to $Q1$ and the beginning of the moment corresponding to $Q2$ and another big intersection exists between the moments corresponding to $Q2$ and $Q3$. From Fig. 10, **MMN** Wang et al. (2022) makes a serious prediction for $Q1$, locating the moment corresponding to $Q2$. Meanwhile, when predicting $Q3$, **MMN** omits the temporal region intersected with $Q2$ but correct temporal region also belonged to the moment of $Q3$ for the final prediction. Compared to **MMN**, our **Only Static** and **Only Dynamic** predict more accurate moments for each query, and they can accurately comprehend the intersection of $Q2$ and $Q3$. **Only Static** performs better at identifying transitions, while **Only Dynamic** performs better at recognizing overlapping events. Our **Full Model** performs best in these challenging scenarios because it combines the advantages of **Only Static** and **Only Dynamic**.

**Video Sample with Similar Scenes.** The frames in video sample "ID: v_UajYunTsr70" from the ActivityNet Captions dataset too have high similarity, as shown in Fig. 10 (b). You can find it to locate the corresponding moment corresponding to $Q1$: "*A cat is sitting on top of a white sheet.*" **MMN** is basically unable to distinguish the video content for the three different queries. It almost predicts the entire video for each query. Even through our **Only Static** performs poorly in this situation too, our **Only Dynamic** performs much better than MMN. Finally, our **Full model** locates the most accurate target moment. This is thanks to our model that combines the advantages of static and dynamic modules, especially for that the latter learns tighter contextual correlation of video in this case.

### C.2   VISUALIZATION ON ACTIVITYNET SPEECH FOR SLVG.

We also provide quantitative results of our UniSDNet on SLVG to demonstrate the effectiveness of our model in the video grounding task based on spoken language.

**Video Sample with Noisy Background.** We instantiate the video sample "ID: v_FsS8cQbfKTQ" from the ActivityNet Speech dataset in Fig. 11 (a) using audio queries under noisy background interference. We can see that **MMN** predicts the video clips corresponding to $Q2$ and $Q3$ with significant deviations, and the predicted moments totally do not intersect with **GT** at all. This video is a challenging case. Compared to **MMN**, **Only Static** and **Only Dynamic** coverage the queried moment but have somewhat boundary shifts. exhibits a strong advantage, as it correctly predicts the relative positions of all events has a large intersection ratio with **GT** video clips. Compared to **MMN**, our **Full model** exhibits the best prediction results for all queries, as it correctly predicts the queried moment and has a large intersection ratio with **GT** video clips. From the 2D map in the figure, it can be seen that our model still performs well in video grounding task based on audio queries, fully demonstrating the generalization of our model.

**The Videos with Continuous and Varied Actions.** Taking video sample "ID: v_UJwWjTvDEpQ" from the ActivityNet Speech dataset in Fig. 11 (b) as an example, the video shows a scene with a clean background, but in which a boy's actions are continuously changing. In this case, for different event divisions, it is necessary to finely distinguish the contentual semantics of the boy's actions and the differences between them. **MMN** fails to recognize such densely varied actions and incorrectly assigns the whole video as the answer (*e.g.*, $Q1$ and $Q2$). **Our Static** predicts the approximate location of each event. **Our Dynamic** exhibits excellent performance in distinguishing the semantics of continuous actions, it not only correctly distinguishes the semantic centers of three events, but also more accurately predicts the boundaries of each event, compared to **MMN** and **Our Static**. Inspiring, **Full Model** achieves the most accurate prediction of the location and semantic boundaries of events, this is thanks to the combination of static and dynamic modes, which deepens the understanding of video context and enables the model to distinguish different action semantics.

### C.3   MORE VISUALIZATION OF PLETHORIC MULTI-QUERY CASES

**Visualization Examples on the TACoS Dataset.** Taking the video sample "ID: s27-d50" in Fig. 12 (a) as an example, we provide the grounding results of our model and MMN. Note that the total duration of the video is 82.11 s, which includes 119 query descriptions. Limited by page size and layout, we select and show 6 very challenging queries here. The video depicts a person cooking in a kitchen. **MMN** experiences a significant prediction error in the moment corresponding to the query $Q88$. In contrast, our **Full model** accurately determines the relative positions of the video segments corresponding to all the queries. The qualitative results highlight the effectiveness of learning semantic associations between multi-queries (*i.e.*, multi-queries contextualization) for cross-modal video grounding.

**Visualization Examples on the Charades-STA Dataset.** The video sample "ID: U5T4M" in Fig. 12 (b) has a duration of 19.58 s, which describes the indoor activities of a person, and contains 7 queries. Our **Full model** infers the localization results of all queries corresponding to the video at once. In all queries, $Q1$ and $Q2$ are similar descriptions of an event, respectively. The same situation also includes queries of $Q4$ and $Q5$, $Q6$ and $Q7$. We can see that our **Full model** accurately predicts the boundaries of each query, and effectively distinguishing the semantics among similar but with slightly different events.

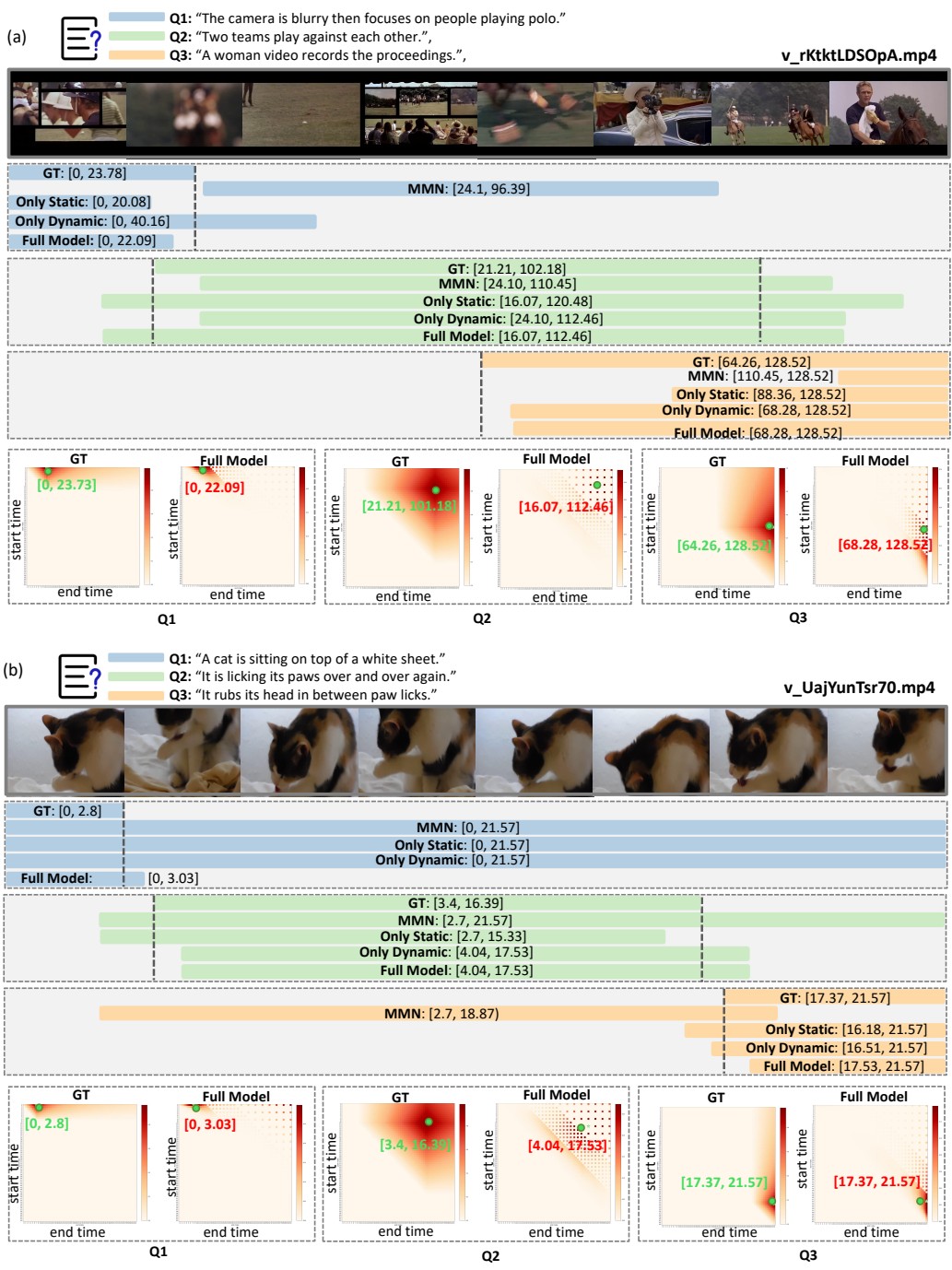

Figure 10: Qualitative examples on ActivityNet Captions for NLVG. **(a)** The video contains complex scene transitions and overlap. **(b)** The video scenes that are difficult to distinguish. **MMN** makes significant errors in predicting the location range of the queried events, *i.e.*, $Q1$ and $Q3$ in cases (a) and (b), respectively. Our **Only Static** has an advantage in predicting transitions ($Q1$ in case (a)), our **Only Dynamic** performs better in predicting overlapping. It is difficult to distinguish scenarios ($Q2$ and $Q3$ in both cases (a) and (b)). Our **Full Model** performs best in both challenging scenarios, as it combines the advantages of static (query semantic differentiation) and dynamic (video sequences context association) modules.

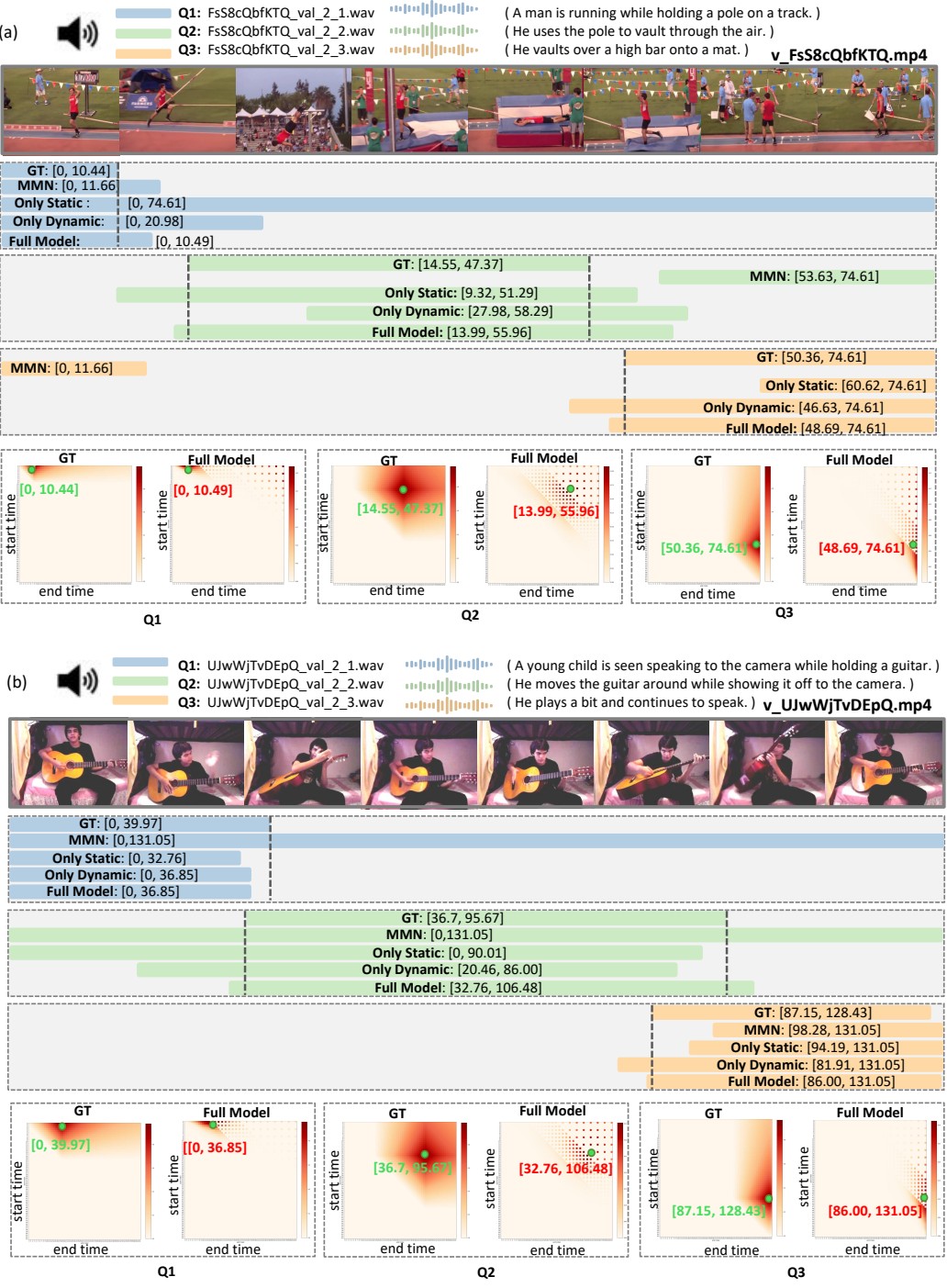

Figure 11: Qualitative examples on ActivityNet Speech for SLVG. **(a)** The scenes that contains a noisy background. **(b)** The Videos with Continuous and Varied Actions. **MMN** makes significant errors in predicting the location ($Q2$ and $Q3$ in case (a)) and location coverage areas of events ($Q1$ and $Q2$ in case (b)). These two cases are challenging. Encouragingly, our **Full Model** achieves the best performance in these video grounding cases based on audio queries, this confirms the effectiveness and generalization of our unified static and dynamic methods in this task.

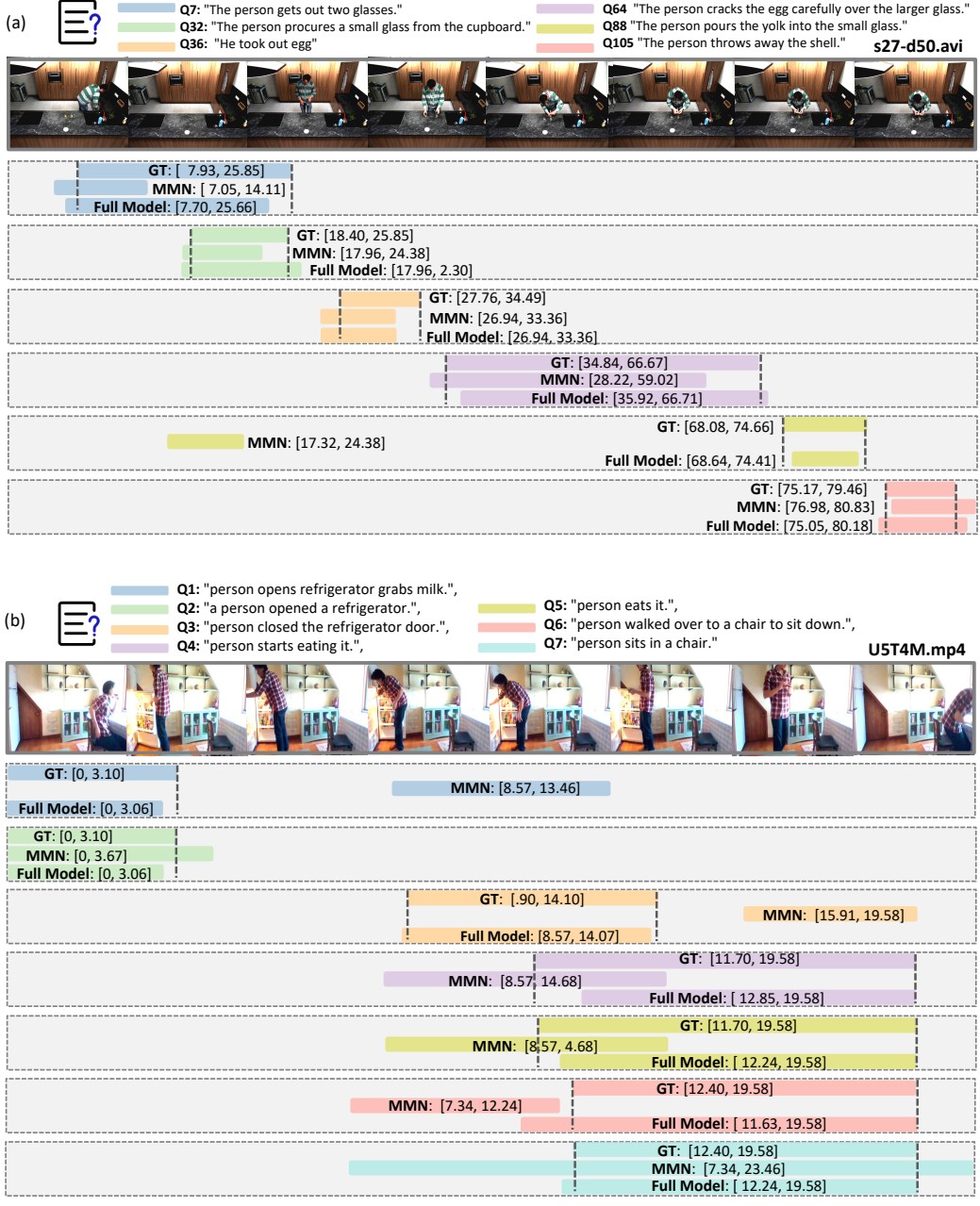

Figure 12: Quantitative examples of plethoric multi-query cases. **(a)** Examples on the TACoS dataset for NLVG. **(b)** Examples on the Charades-STA dataste for NLVG. **MMN** has a significant semantic bias when predicting $Q7$ in case (a). and $Q4, Q5, Q7$ in case (b), there is also a big positional deviation in predicting $Q88$ in case (a), and $Q1, Q3$ in case (b). Our **Full Model** correctly predicts the location of all the queried events, and the predicted moment interval is closest to that of **GT**, this is thanks to model capacity of mutual learning of video and multiple queries and effectively capturing the video context associated with multiple queries.

