# OpenReview forum: "Unified Static and Dynamic: Temporal Filtering Network for Efficient Video Grounding"
_ICLR.cc/2024/Conference — Submitted to ICLR 2024_

### Official Review · Reviewer_GE2s · 2023-11-01

**Soundness:** 3 good
**Presentation:** 4 excellent
**Contribution:** 3 good
**Rating:** 8
**Confidence:** 4

**Summary:**

This paper tackles two video grounding problems based on natural and spoken language queries. Inspired by the human biology, the authors propose a novel framework, called UniSDNet, that enables both static and dynamic interactions to facilitate the learning of video grounding. The static interaction is implanted by a series of ResMLP layers, while the dynamic interaction is conducted by graph convolution with Gaussian Radial filters. Experimental results on three benchmarks for each task confirm the effectiveness of the proposed framework.

**Strengths:**

+ The manuscript is overall well-organized and easy to follow.
+ The motivation behind the static and dynamic interactions based on the brain activity is clear and compelling.
+ The two-stage information aggregation methods are shown to be effective, where each component is appropriately designed.
+ The experimental results are very strong, clearly outperforming the existing approaches for both grounding tasks.
+ The collected spoken language grounding datasets will significantly benefit the research community. The authors are encouraged to publish the code and data after the review process.

**Weaknesses:**

I did not find major weaknesses in this paper, yet summarize some questions about the method below.

- What is the motivation behind the implementation of Static Semantic Supplement Network? I am wondering how the cross-modal interaction is performed through the MLP layers. To my understanding, the shared weights across different modalities would extract some common features spanning different modalities. Some analytical experiments on this would be beneficial. Also, the architecture design seems similar to that of Transformer blocks except for the self-attention. What happens if we use the conventional Transformer layers?
- The proposed architecture exploits multiple queries at once, to facilitate the model learning. However, how the number of queries affects the performance is not diagnosed. An ablative study on the number of queries regarding performance and cost would be helpful.
- In Figure 5, the effectiveness of the proposed filtering GCN is clearly verified. On the other hand, there are some interesting tendency differences between NLVG and SLVG. That is, the graph convolution layer itself is important, yet different layer modeling brings insignificant performance gaps on NLVG. In contrast, on SLVG, the graph modeling brings negligible gains alone, but the proposed filtering mechanism shows substantial improvements. How can one interpret this phenomenon? If you have, please share some insights.
- The proposed method is well validated in the datasets with one-to-one matching between queries and moments. How would it perform for one-to-many matching datasets, such as QVHighlights [1]?

[1] Lei et al. "QVHighlights: Detecting Moments and Highlights in Videos via Natural Language Queries", Neurips, 2021.

(Minor)

The manuscript contains some formatting errors due to the excessively small margins between captions and the main text. They should be handled appropriately to raise the quality of the paper.

**Questions:**

Please refer to the Weakness section.

---

> ### Author Response · Authors · 2023-11-21
> **The Response to Reviewer GE2s (Part 1/3)**
>
> We are so encouraged and cheerful to receive your constructive comments, your positive reception of UniSDNet's biological motivation and recognition of our two-stage information aggregation methods are both encouraging and deeply appreciated.
>
> > **Q1: What is the motivation behind the implementation of Static Semantic Supplement Network? I am wondering how the cross-modal interaction is performed through the MLP layers. To my understanding, the shared weights across different modalities would extract some common features spanning different modalities. Some analytical experiments on this would be beneficial. Also, the architecture design seems similar to that of Transformer blocks except for the self-attention. What happens if we use the conventional Transformer layers?**
>
> **A1:**
> Thank you for raising the concerns about the motivation of static network. We would like to explain the implementation of our static network from the perspectives of motivation, technology, and experiment.
>
> - **Motivation-wise,** the implementation of Static Semantic Supplement Network is strongly inspired by the **global neural workspace (GNW)** theory in biology [R15], that is, for video understanding, the brain will perform 'global broadcast communication' (interaction) of the multi-source information in the early stage of visual perception for the video, while the cognitive consciousness of the brain will **strengthen perceptual conditions** and **delay conditioned reflexes**, enabling the human brain to understand new conditioned reflexes brought about by perceptual information updates.
>
>     On the premise of the above theory, this inspires us to utilize residual networks structure to update our understanding of video sequences in the delaying conditioned reflex states by aggregating the perceptual information. **Multilayer perceptron (MLP)** is proven to be effective and has strong interpretability in modeling multi-level cognitive processing [R19], thus we use MLP to aggregate the perceptual information. The visual consciousness process is modeled as follows：$x^{n+1}$ = $\text{ResMLP}(x^n)$ = $x^n$+$\text{MLP}(x^n)$, where $x^n$ represents the perception condition in the current state, $x^{n+1}$ denotes the reflection result, and $\text{ResMLP}(x^n)$ can enhance the information of the perception condition $x^n$.
>
> - **Technology-wise**,  we concatenate multiple queries and video clips together as a sequence input into the network for fully connected interaction of all tokens, its main function is to supplement and associate semantics between modalities before the latter graph filtering, provide more video descriptions information and significantly fill the gap between vision-language modalities, aiding in understanding video content.
>
> - **Experiment-wise**, as per your valuable suggestion, we have tested the effect of Transformer as a static network as shown in the table below (**Table R6**), and as you are concerned, Reviewer 'TT77' has also focused on the ablation study of the static module. From the results, in terms of performance and efficiency, Transformer is close to our method, but our results are better. We speculate that the reason is that our network also includes the second stage of graph filtering. The static network uses a lightweight and stable network, which is more conducive to model training. Using Transformer as a static network increases the weight and instability factors [R20] of the network.
>
> **Table R6: Results of different static networks on the ActivityNet Captions.**
>
> |Static | Infer. Speed (s/query) | R@1, IoU@0.3 | R@1, IoU@0.5 | R@1, IoU@0.7  |  R@5, IoU@0.3 | R@5, IoU@0.5  | R@5, IoU@0.7 | mIoU  |
> | :---------- | :----------: | :----------:  | :----------: | :----------:  | :----------:  | :----------:  | :----------:  | :----------: |
> | Transformer | 0.024 | 75.17 | 59.98 | 38.38 | 91.26 | 85.77 | 74.25 | 54.97|
> |**Our ResMLP** | **0.009** | **75.85**	| **60.75**	| **38.88**	| **91.16** | **85.34** | **74.01**	| **55.47** |
>
> **References:**
>
> [R15] Volzhenin et al., Multilevel development of cognitive abilities in an artificial neural network. Proceedings of the National Academy of Sciences, 119(39), pp. e2201304119, 2022.
>
> [R19] Chavlis et al., Drawing inspiration from biological dendrites to empower artificial neural networks. Current opinion in neurobiology. 70, pp. 1-10, 2021.
>
> [R20] Hugo et al., Resmlp: Feedforward networks for image classification with data-efficient training. TPAMI, 45(4):5314–5321, 2022.

---

> > ### Author Response · Authors · 2023-11-21
> > **The Response to Reviewer GE2s (Part 2/3)**
> >
> > > **Q2: The proposed architecture exploits multiple queries at once, to facilitate the model learning. However, how the number of queries affects the performance is not diagnosed. An ablative study on the number of queries regarding performance and cost would be helpful.**
> >
> > **A2:**
> > Thanks for your kind and valuable suggestion. Following your suggestions, we have conducted ablation studies on the impact of query quantity on model performance on the ActivityNet Captions dataset. An explanation of the distribution of query quantity per video is that, the training set of this dataset contains 10,009 videos, with a minimum of 2 and a maximum of 27 queries per video. Due to the number of queries for each video in this dataset being concentrated in [3, 8], we set the upper limits for the number of queries $M$ per video feed as 1, 3, 5, 8, and no upper limit. In fact, the 25\%, 50\%, and 75\% quantiles are 3, 3, and 4, respectively. When the $M$ = 8, 97.33\% of samples in the dataset are equipped with full queries.
> >
> > The results are shown in **Table R7** below, and according to the experiments, when the number of training query inputs $M=1$, our model is comparable with state-of-arts. As the query number upper limit increases, the performance of our model improves, which demonstrates the effectiveness of our model in utilizing multimodal information.
> >
> > **Table R7: The impact of query quantity on ActivityNet Captions dataset for NLVG task.**
> >
> > |Method (\#Query) | R@1, IoU@0.3 | R@1, IoU@0.5 | R@1, IoU@0.7  |  R@5, IoU@0.3 | R@5, IoU@0.5  | R@5, IoU@0.7 | mIoU  |
> > | :---------- | :----------:  | :----------:  | :----------:  | :----------:  | :----------:  | :----------:  | :----------: |
> > | UniSDNet ($M=1$)     | 68.66 | 52.35 | 32.25 | 89.74 | 83.35 | 70.61 | 50.22 |
> > | UniSDNet ($M\leq 3$) | 72.42 | 57.30 | 36.64 | 89.82 | 83.88 | 72.14 | 53.30 |
> > | UniSDNet ($M\leq 5$) | 74.41 | 59.27 | 37.70 | 90.83 | 85.07 | 74.17 | 54.58 |
> > | UniSDNet ($M\leq 8$) | 74.79 | 60.28 | 38.35 | 90.68 | 85.15 | 73.90 | 55.03 |
> > | **UniSDNet (Full)**  |**75.85**| **60.75** | **38.88** | **91.16** | **85.34** | **74.01** | **55.47** |
> >
> > > **Q3: In Figure 5, the effectiveness of the proposed filtering GCN is clearly verified. On the other hand, there are some interesting tendency differences between NLVG and SLVG. That is, the graph convolution layer itself is important, yet different layer modeling brings insignificant performance gaps on NLVG. In contrast, on SLVG, the graph modeling brings negligible gains alone, but the proposed filtering mechanism shows substantial improvements. How can one interpret this phenomenon? If you have, please share some insights.**
> >
> > **A3:**
> > Thank you for your detailed comments. Compared to other graph methods, our dynamic model exhibits interesting inconsistencies in NLVG and SLVG tasks. We hope to provide our own humble opinion on this phenomenon.
> > We infer that a large part of this is due to the different feature characteristics of text and audio.
> > In our experiments, we use DistilBERT as textual features and Data2vec as audio features, where DistilBERT is an acknowledged high-quality textual features.
> > For the NLVG task, the performance improvement with high-quality features brought by the method development is approaching saturation.
> > In this case, if the R@1, IoU@0.7 indicator increases by 1\%, it indeed means a significant improvement for NLVG.
> > As for SLVG, current works are relatively rare, and the audio-video interaction has the potential to be developed deeply.

---

> > > ### Author Response · Authors · 2023-11-21
> > > **The Response to Reviewer GE2s (Part 3/3)**
> > >
> > > > **Q4: The proposed method is well validated in the datasets with one-to-one matching between queries and moments. How would it perform for one-to-many matching datasets, such as QVHighlights [R21]?**
> > >
> > > **A4:**
> > > Thanks for pointing out the excellent work of QVHighlights and constructive comments.
> > > QVHighlights is a recently publicized dataset for both moment retrieval (MR) and highlight detection (HD) tasks.
> > > Following the practice [R21, R22], the commonly used metric is Recall@K, IoU=[0.5, 0.7].
> > > As your valuable suggestion, we have tested our model on this dataset. Current works on the QVHighlights dataset have undergone multi-task learning, including both MR and HD tasks.
> > > However, due to the urgent rebuttal deadline, we only complete our model evaluation for the MR single task mode on this dataset.
> > > We have listed the reported results of exiting works for the MR single task in **Table R8** and compared ours with theirs. From the results and facts, there are some conclusions.
> > >
> > > - Our method still performs the best with signle MR task test in one-to-many query matching mode.
> > > This also proves that the good universality of our model for the task setup of one query to multiple moments retrieval, in which each query corresponds to multiple moments retrieval.
> > >
> > > - To be applicable to both HD and MR tasks, M-DETR [R21] and UMT [R22] are extended on the basis of the DETR method with the learnable moment queries. We adopt the typical 2D temporal proposal infrastructure and our methods still performs well in solving one-to-many matching MR task. This demonstrates our model's advantage of high precision (R@1, IoU@0.7 is 40.39, much higher than the 33.82 of UMT, and 25.87 of M-DETR).
> > >
> > > - In the future, we will continue to collaborate with HD tasks, whose multi-task joint training method has been proven to further promote the MR results [R22],
> > > and will finish this comparison with fair conditions and believe that we will have better performance.
> > >
> > > **Table R8: Comparison with the state-of-art for one-to-many query matching on QVHighlights for single MR task.**
> > > | Method       |        Venue | Task | Video | R@1, IoU@0.5 | R@1, IoU@0.7  |
> > > | :---------- | :----------  | :----------:  | :----------:  | :----------:  | :----------:  |
> > > | M-DETR [R21] | NeurIPS 2021 | Single MR task | Slowfast + CLIP | 44.84   |    25.87    |
> > > | UMT [R22]    | CVPR 2022    | Single MR task | Slowfast + CLIP | 54.14   |    33.82    |
> > > | **Ours**     |       | **Single MR task** | **Slowfast + CLIP** | **54.58**   |    **40.39**   |
> > >
> > > **References:**
> > >
> > > [R21] Lei et al., QVHighlights: Detecting Moments and Highlights in Videos via Natural Language Queries. NeurIPS, pp. 11846--11858, 2021.
> > >
> > > [R22] Liu et al., Umt: Unified multi-modal transformers for joint video moment retrieval and highlight detection. CVPR, pp. 3042-3051, 2022.
> > >
> > >
> > > > **Q5: The manuscript contains some formatting errors due to the excessively small margins between captions and the main text.**
> > >
> > > **A5:** Thank you for pointing out the format issues. We have carefully proofread our paper and fixed them.
> > >
> > > We hope our response addresses your concerns. If there are any further questions, please do not hesitate to let us know. Thank you for your time and valuable feedback.

---

> ### Comment · Reviewer_GE2s · 2023-11-23
> **Post-rebuttal Comment**
>
> Thanks to the authors for the careful response.
>
> I believe most of my concerns are appropriately addressed.
>
> Especially, the ablative experiments regarding the number of queries as well as the results on QVHighlights are substantially valuable.
>
> I will keep my rating unchanged with the hope that the authors will reflect on the discussion made during the rebuttal in the manuscript.

---

### Official Review · Reviewer_Z5of · 2023-11-04

**Soundness:** 2 fair
**Presentation:** 2 fair
**Contribution:** 2 fair
**Rating:** 5
**Confidence:** 3

**Summary:**

The paper proposes a new approach to natural language-based (NLVG) and spoken language-based video grounding (SLVG). It first uses a MLP with residual connection to model the interaction between video feature and queries. Next, it proposes a graph network to model the short-term dynamics. The proposed model achieves great improvements on both NLVG and SLVG benchmarks and runs faster than the multi-query benchmark.

**Strengths:**

1. Good performance on both NLVG and SLVG benchmarks.

2. It is nice to see an extension from NLVG to SLVG with a newly proposed benchmark. The proposed method proves effective on both tasks.

3. Detailed implementation details and prediction analysis in the appendix.

**Weaknesses:**

1. The inspiration from human visual perception biology is not very motivating. Specifically, it is hard to see why a MLP with residual connection is the way to achieve the “global broadcast communication” of the brain. Either bridge the gap or Simply drop the bio-inspiration and go straight into the technical method.

2. When expanding a single gaussian kernel to multi-kernel Gaussian, it seems that only the bias $z_i$ is sweeping? Have you tried different $\gamma$?

3. Ablation in Fig 5 shows mostly similar results especially on NLVG, indicating that the designs in Dynamic Filter Graph actually do not quite matter.

**Questions:**

1. Template

(1) The first page is missing a header.

(2) Please change `\cite{..}` to `\citep{..}` for clarity.

---

> ### Author Response · Authors · 2023-11-21
> **The Response to Reviewer Z5of (Part 1/3)**
>
> Thank you very much for your recognition of our contributions to NLVG and SLVG tasks. Your constructive comments are exceedingly helpful for us to improve our paper.
>
> > **Q1: The inspiration from human visual perception biology is not very motivating. Specifically, it is hard to see why a MLP with residual connection is the way to achieve the 'global broadcast communication' of the brain. Either bridge the gap or Simply drop the bio-inspiration and go straight into the technical method.**
>
> **A1:**
> Thank you very much for your attention to the design motivation of static networks (MLP with residual connection, ResMLP). Here, we clarify the motivation to bridge the gap between the biological theory 'global broadcast communication' and the technical design of static network ResMLP.
> Sincerely, our work has been strongly inspired by the latest theories of human visual perception [R14, R15]. The **global neuronal workspace (GNW)** theory mentioned in [R15] is the foundation of conscious work in visual understanding [R15, R16, R17], it can achieve the fusion process of multi-source information in the early stage of visual event recognition, that is, the **"global broadcast communication"** of the brain. GNW theory indicates that cognitive consciousness has the function of **strengthening perceptual conditions** and **delaying conditioned reflexes** [R15, R18], enabling the human brain to understand new conditioned reflexes brought about by perceptual information updates [R15, R16].
>
> On the premise of the above theory, this inspires us to utilize residual network structure to update our understanding of video sequences in the delaying conditioned reflex states by aggregating the perceptual information. **Multilayer perceptron (MLP)** is proven to be effective and has strong interpretability in modeling multi-level cognitive processing [R19], thus we use MLP to aggregate the perceptual information. The visual consciousness process is modeled as follows: $x^{n+1} = \text{ResMLP}(x^n) = x^n+\text{MLP}(x^n),$ where $x^n$ represents the perception condition in the current state, $x^{n+1}$ denotes the reflection result, and $\text{ResMLP}(x^n)$ can enhance the information of the perception condition $x^n$.
>
> Technically, in this work, we adopt a unified framework of static (ResMLP) and dynamic (graph filtering) structures for video grounding. In the early stage, this global broadcast mode of ResMLP first perceives all multimodal information, while the latter graph filter purifies key visual information.
> This pre-broadcast learning mode of ResMLP is very necessary as a global biological perception mechanism and has also verified its effectiveness via extensive experiments, as discussed in the paper. We will add the explanation on the rationality of our static network design from a **technical perspective** and incorporate your suggestion into the paper writing by considering both motivation and technical implements for better clarification.
>
> **References:**
>
> [R14] Barbosa et al., Interplay between persistent activity and activity-silent dynamics in the prefrontal cortex underlies serial biases in working memory. Nature neuroscience, 23(8), pp. 1016-1024, 2020.
>
> [R15] Volzhenin et al., Multilevel development of cognitive abilities in an artificial neural network. Proceedings of the National Academy of Sciences, 119(39), pp. e2201304119, 2022.
>
> [R16] Cleeremans et al., Consciousness matters: Phenomenal experience has functional value. Neuroscience of consciousness. 2022, pp. niac007, 2022.
>
> [R17] Richards et al., A deep learning framework for neuroscience. Nature neuroscience, 22, pp. 1761-1770, 2019.
>
> [R18] Grover et al., Differential mechanisms underlie trace and delay conditioning in Drosophila. Nature, 603, pp. 302-308, 2022.
>
> [R19] Chavlis et al., Drawing inspiration from biological dendrites to empower artificial neural networks. Current opinion in neurobiology. 70, pp. 1-10, 2021.

---

> > ### Author Response · Authors · 2023-11-21
> > **The Response to Reviewer Z5of (Part 2/3)**
> >
> > > **Q2: When expanding a single gaussian kernel to multi-kernel Gaussian, it seems that only the bias z is sweeping? Have you tried different $\gamma$?**
> >
> > **A2:** Thank you for the valuable feedback.
> > Considering to better model the continuity and complexity of visual perception transmission, **a single-kernel Gaussian filter** is insufficient to reflect the comprehensive degree of event associations in the video [R14]. We adopt **multi-kernel Gaussian filters** as a rich **Filter-generating-networks** architecture to extend the scalar $d_{ij}$ to high-dimensional space to achieve the goal.
> > In this work, we employ the multi-kernel Gaussian $\phi_k(x)=exp(-\gamma(x-z_k)^2),  k \in [1, h]$, and there are three variables ($z_k,h,\gamma$): different bias $\{z_k\}$ for total $h$ Gaussian kernels and a Gaussian coefficient $\gamma$, where $z_k$ is a bias to avoid a plateau at the beginning of training due to the highly correlated Gaussian filters.
> > To meet the constraint of nonlinear correlated Gaussian kernels, we randomly set biases $z_k$ at equal intervals (e.g., 0.1 or 0.2) starting from 0.0, sweep the value of $h$ from 25 to 200 (see the ablation experiment Table 9 in Appendix and the best setting is $h=50$) and set the global range of $\{z_k\}$ values to $[0, 5]$ in our experiments.
> >
> > As per your valuable suggestion, we add new experiments about different $\gamma$ in **Table R4**. Gaussian coefficient $\gamma$ reflects the amplitude of Gaussian kernel function that controls the gradient descent speed of the function value.
> > It can be find that from **Table R4**, when $\gamma=25.0$, our model achieves the best performance (in bold font of **Table R4**). We also list the average and standard deviation of the five experimental results of $\gamma=\{5.0, 10.0, 25.0, 50.0, 70.0\}$.
> > We select $\gamma=10.0$ as the empirical setting as its result (with *italic* in **Table R4**) is closest to the average *avg.*. To summarize, in our experiments, the final settings of variables  ($h,\gamma$) are set to 50 and 10.0, and $\{z_k\}$ is set at an equal interval of 0.1.
> > We will add this new ablation and more explanation to the paper.
> >
> >
> > **Table R4: The ablation study about different Gaussian coefficients $\gamma$ on the ActivityNet Captions.**
> >
> > |$\gamma$ | R@1, IoU@0.3 | R@1, IoU@0.5 | R@1, IoU@0.7  |  R@5, IoU@0.3 | R@5, IoU@0.5  | R@5, IoU@0.7 | mIoU  |
> > | :----------: | :----------: | :----------: | :----------: | :----------: | :----------: | :----------: | :----------: |
> > |  5.0 | 75.76  | 60.80 | 39.23 | 91.14 | 85.43 | 74.33 | 55.51 |
> > | *10.0* | *75.85*  | *60.75* | *38.88* | *91.16* | *85.34* | *74.01* | *55.47* |
> > | **25.0** | **75.87**  | **60.77** | **39.30** | **91.16** | **85.23** | **74.06** | **55.52** |
> > | 50.0 | 75.84  | 60.98 | 38.83 | 91.04 | 85.27 | 73.98 | 55.51 |
> > | 75.0 | 75.74  | 60.57 | 38.63 | 90.98 | 85.26 | 73.86 | 55.29 |
> > | avg. | 75.81  | 60.77 | 38.97 | 91.10 | 85.31 | 74.05 | 55.46 |
> > | std. | 0.06   |  0.15 |  0.28 | 0.08  |  0.08 | 0.17  | 0.10 |
> >
> >
> > > **Q3: Ablation in Fig 5 shows mostly similar results especially on NLVG, indicating that the designs in Dynamic Filter Graph actually do not quite matter.**
> >
> > **A3:**
> > Thank you for your valuable feedback. Fig 5 in the main paper may cause your misunderstanding. Please allow us to provide additional results below (**Table R5**) to address your concerns.
> > As shown in **Table R5**, our DTFNet consistently achieves the best performance compared to other graph settings, such as achieving 38.88 at the metric R@1, IoU@0.7.
> >
> > *And, why are the results of these graph methods relatively close for NLVG task not as well as the performance improvement of SLVG?*
> > There may be different characterizations of original textural and audio features.
> > In our experiments, we use DistilBERT as textual features and Data2vec as audio features, where DistilBERT is an acknowledged high-quality textual features.
> > For the NLVG task, the performance improvement with high-quality features brought by the method development is approaching saturation.
> > In this case, while the R@1, IoU@0.3 increases by 2.36% and the mIoU increases by 1.92% compared to GAT, they indeed mean a significant improvement for NLVG.
> > As for SLVG, the audio-video interaction has the potential to be developed deeply.
> >
> > **Table R5: Different dynamic modeling methods on the ActivityNet Captions for NLVG task.**
> >
> > |Method | R@1, IoU@0.3 | R@1, IoU@0.5 | R@1, IoU@0.7  |  R@5, IoU@0.3 | R@5, IoU@0.5  | R@5, IoU@0.7 | mIoU  |
> > | :---------- | :----------: | :----------: | :----------: | :----------: | :----------: | :----------: | :----------: |
> > | GCN | 73.33  | 58.14 | 38.57 | 89.86 | 84.52 | 73.22 | 53.53 |
> > | GAT | 73.49  | 58.78 | 38.32 | 89.92 | 84.60 | 72.21 | 53.55 |
> > | D   | 74.40  | 59.16 | 38.46 | 90.42 | 84.62 | 73.26 | 54.06 |
> > | MLP | 74.29  | 59.52 | 38.07 | 90.52 | 84.48 | 72.88 | 54.12 |
> > | **Our DTFNet** | **75.85**  | **60.75** | **38.88** | **91.16** | **85.34** | **74.01** | **55.47** |

---

> > > ### Author Response · Authors · 2023-11-21
> > > **The Response to Reviewer Z5of (Part 3/3)**
> > >
> > > > **Q4: Template: (1) The first page is missing a header. (2) Please change  \\cite{..} to \\citep{..} for clarity.**
> > >
> > > **A4:**
> > > Thank you for pointing out the format issues. We have carefully proofread our paper and fixed them.
> > >
> > > We hope our response addresses your concerns. If there are any further questions, please do not hesitate to let us know. Thank you for your time and valuable feedback.

---

### Official Review · Reviewer_TT77 · 2023-11-04

**Soundness:** 3 good
**Presentation:** 3 good
**Contribution:** 3 good
**Rating:** 6
**Confidence:** 4

**Summary:**

To address the Natural Language Video Grounding (NLVG) and Spoken Language Video Grounding (SLVG) problems, this paper introduces a Unified Static and Dynamic Network (UniSDNet). In which, the Static Network utilizes ResMLP layer to model global context while the Dynamic Network leverages the multi-kernel Temporal Gaussian Filter to build graph, where the gaussian filter leverages the temporal distance and semantic similarity. Extensive experiments demonstrate promising results.

**Strengths:**

1. Most parts of the paper is well-written, clearly demonstrating the motivation,  methodology and experiments. The methodology part is kind of easy to follow.
2. The idea is motivated from the human visual perception biology, which formulates an interesting story for this paper.
3. Extensive experiments successfully demonstrate the effectiveness of each proposed component of this work, which is good.
4. The visualization and figures are plus to show more intuitions.
5.  The final results of this paper achieves the state-of-the-art from both efficiency and effectiveness perspectives.

**Weaknesses:**

1. The introduction reads like a related work. It will be great to make more comparison between this work and previous work. Answering what is wrong with previous works? and where the efficiency and performance gain come from in this paper?
2. This paper introduces some new/confusing terminologies with their own definition, which hurts the reading experience. For example, 'static semantic supplement network' and 'activity-silent mechanism' are actually the global context interaction.
3. Although the motivation of static and dynamic network is demonstrated, the justification of specific design is not enough. For example, in the static network, transformer architecture or the recent S4[1] architecture can also be used as long-range filter. Some ablation studies regarding either the performance or efficiency would be great to include.
4. In the dynamic network, not sure why use Gaussian filter on the distance (d_{ij}). Can you provide more insights? why not directly use the distance.
5. No notation for the 'FNN'. Is this the feedforward network?
6. In the Figure 5, no notation/description for 'D'.

[1]  Efficiently modeling long sequences with structured state spaces. ICLR 2021

**Questions:**

Is there any chance also leverage the audio signal into this work, formulating a multi-model graph?

---

> ### Author Response · Authors · 2023-11-21
> **The Response to Reviewer TT77 (Part 1/3)**
>
> Thanks for your insightful and constructive feedback on our manuscript. Your positive reception of our visual perception biological motivation to guide network design and recognition of our method and experiments are both encouraging and deeply appreciated. We will make some clarifications in accordance with your suggestions.
>
>
> > **Q1: The introduction reads like a related work. It will be great to make more comparison between this work and previous work. Answering what is wrong with previous works? and where the efficiency and performance gain come from in this paper?**
>
> **A1:** Thanks for your valuable suggestion. Technology-wise, previous methods mostly focus on solving a certain aspect of the Temporal Video Grounding (TVG) task, such as representation learning of language and video self-modality [R1, R2], multimodal fusion [R3, R4], cross-modal interaction [R5, R6], proposal candidate generation [R7, R8], proposal-based cross-modal matching [R9, R10], target moment boundary regression [R11, R12], and so on.
> Our work actually proposes a new paradigm to establish a two-stage unified static and dynamic semantic complementary new architecture. Its unique characteristics are that
> - Processing multimodal signals in a unified ResMLP network, while many previous works are independently encoding the language modality and video modality [R1-R7, R9-R12].
> - After the implementation of the above static ResMLP, we introduce a Gaussian nonlinear filtering method to learn the semantic associations within the video and combine it with the back-end proposal generation to promote cross-modal semantic alignment, further developing the proposal-based TVG method.
>
> Our work is inspired by visual perception biology. This unified static and dynamic two-stage architecture performs excellent joint learning of language and video, which achieves state-of-the-art performance on NLVG and SLVG tasks.
> The ablation experiments in **Table 4** of our manuscript demonstrate the effectiveness of each proposed component of this work. We will improve our manuscript based on your constructive suggestions.
>
> **Eﬀiciency and performance:** Technology-wise, the inference efficiency of our model comes from a streamlined architecture design, which allows for parallel inference of video and multiple queries in a unified ResMLP, saving time and expenses.
> Our promising model performance mainly comes from our model's comprehensive information interaction with query and video (ResMLP, static network), as well as our detailed inference of video content (DTFNet, dynamic graph filtering network), which was confirmed in the first ablation experiment in **Table 4** of the manuscript.
>
> **References:**
>
> [R1] Xia et al., Video-guided curriculum learning for spoken video grounding. ACM MM, pp. 5191–5200, 2022.
>
> [R2] Rodriguez et al., Memory-efficient temporal moment localization in long videos. ECAL, pp. 1901–1916, 2023.
>
> [R3] Li et al., Proposal-free video grounding with contextual pyramid network. AAAI, pp, 1902–1910, 2021.
>
> [R4] Liu et al., Exploring optical-flow-guided motion and detection-based appearance for temporal sentence grounding. IEEE TMM, 2023.
>
> [R5] Liu et al., Jointly cross-and self-modal graph attention network for query-based moment localization. ACM MM, pp. 4070–4078, 2020.
>
> [R6] Sun et al., Video moment retrieval via comprehensive relation-aware network. IEEE TCSVT, 2023.
>
> [R7] Zhang et al., Learning 2d temporal adjacent networks for moment localization with natural language. AAAI, pp. 12870–12877, 2020.
>
> [R8] Zhang et al., Multi-stage aggregated transformer network for temporal language localization in videos. CVPR, pp. 12669–12678, 2021.
>
> [R9] Gao et al., Fast video moment retrieval. ICCV, pp. 1523–1532, 2021.
>
> [R10] Zheng et al., Phrase-level temporal relationship mining for temporal sentence localization. AAAI, pp. 3669–3677, 2023.
>
> [R11] Zhang et al., Natural language video localization: A revisit in span-based question answering framework. IEEE TPAMI, 44(8), pp. 4252–4266, 2021.
>
> [R12] Liu et al., Skimming, locating, then perusing: A human-like framework for natural language video localization. ACM MM, pp. 4536–4545, 2022.

---

> > ### Author Response · Authors · 2023-11-21
> > **The Response to Reviewer TT77 (Part 2/3)**
> >
> > > **Q2: This paper introduces some new/confusing terminologies with their own definition, which hurts the reading experience. For example, 'static semantic supplement network' and 'activity-silent mechanism' are actually the global context interaction.**
> >
> > **A2:** Thanks for your friendly reminder. To clear up your confusion, we clarify and revise some confusing representations as below.
> >
> > **1) 'static semantic supplement network':** This is the network naming in terms of function, because in our work, we focus on understanding video content in a multimodal environment [R13], and we adopt a unified framework of static and dynamic structures. In the early stage, this global interaction mode first perceives all multimodal information, and then information filtering is performed. In terms of the functionality of the static network for video understanding, it provides more video descriptions information and significantly fills the gap between vision-language modalities, aiding in understanding video content.
> >
> > **2) 'activity-silent mechanism':** This is the mechanism by which the brain processes information in the early stages of human visual perception for the video, as mentioned in [R14]. It manifests as a static multi-source information interaction, achieving the 'global broadcast communication' of the brain. Because we are strongly inspired by this mechanism in static network design, we specifically mentioned this professional biological term in our manuscript. This network can achieve the effect of 'the global context interaction' from a technical perspective, and this is the basic intention for our network design.
> >
> > We will incorporate your suggestion and consider both motivation and technical introduction in the paper.
> >
> > **References:**
> >
> > [R13] Lisa et al., Localizing moments in video with natural language. CVPR, pp. 5803-5812, 2017.
> >
> > [R14] Barbosa et al., Interplay between persistent activity and activity-silent dynamics in the prefrontal cortex underlies serial biases in working memory. Nature neuroscience, 23(8), 1016-1024, 2020.
> >
> >
> > > **Q3: Although the motivation of static and dynamic network is demonstrated, the justification of specific design is not enough. For example, in the static network, transformer architecture or the recent S4[1] architecture can also be used as long-range filter. Some ablation studies regarding either the performance or efficiency would be great to include.
> > [1] Efficiently modeling long sequences with structured state spaces. ICLR 2021.**
> >
> >
> > **A3:**
> > Thank you for your valuable suggestion.
> > For static perception modeling of multi-source information, based on the theoretical guidance of biology, we consider using Multilayer perceptron (MLP) to handle language and video uniformly.
> > The reviewer suggested we consider more long-range filters in our studies, therefore, we have added the ablation experiments about our static network, and the results are shown in **Table R3**:
> >
> > **Table R3: Comparison of different static networks on ActivityNet Captions for NLVG task.**
> >
> > | Static | Infer. Speed (s/query) | R@1,IoU@0.3 | R@1,IoU@0.5 | R@1,IoU@0.7  |  R@5,IoU@0.3 | R@5,IoU@0.5  | R@5,IoU@0.7 | mIoU  |
> > | :---------- | :----------: | :----------: | :----------: | :----------: | :----------: | :----------: | :----------: | :----------: |
> > | Transformer | 0.024 | 75.17 | 59.98 | 38.38 | 91.26 | 85.77 | 74.25 | 54.97 |
> > | S4          | 0.030 | 70.41 | 55.11 | 34.93 | 89.12 | 83.16 | 70.54 | 51.40 |
> > | **Our ResMLP** | **0.009** | **75.85** | **60.75** | **38.88** | **91.16** | **85.34** | **74.01** | **55.47**|
> >
> > The results indicate that:
> > - ResMLP achieves model performance/efficiency trade-offs.
> > - Note that the S4 model is particularly skilled in handling very long (about 1w) sequence tasks, but in the ActivityNet Captions dataset of NLVG tasks, the total number of video clips and queries does not exceed 100, which may be the reason why it is not suitable for our task.
> > - The performance of Transformer is close to ResMLP, but its 'Infer. Speed' cost is 2.67 times that of ResMLP. In terms of better performance and efficiency, we choose ResMLP as the implementation of the static network.
> >
> >
> > > **Q4: No notation for the 'FNN'. Is this the feedforward network?**
> >
> > **A4:** Yes, it is. 'FNN' is the abbreviation of feedforward network. We have unified the expression of FNN and added a full name explanation in the paper.
> >
> >
> > > **Q5: In the Figure 5, no notation/description for 'D'.**
> >
> > **A5:**
> > The following is a description of **D**, and we will provide a clearer explanation of **D** in the paper.
> > In the setting of **D**, we use the message aggregation wight $f_{ij}=1/(d_{ij}+1)$ to replace $f_{ij}$=$\mathcal{F}\_{filter}(d_{ij})$, which indicates that we still consider the clue of temporal distance $d_{ij}$ between two nodes but remove the entire Gaussian filtering calculation from our method.
> > This replacement results in a decrease of 0.42 and 1.92 on R@1, IoU@0.7 for NLVG and SLVG, respectively.

---

> > > ### Author Response · Authors · 2023-11-21
> > > **The Response to Reviewer TT77 (Part 3/3)**
> > >
> > > > **Q6: In the dynamic network, not sure why use Gaussian filter on the distance ($d_{ij}$). Can you provide more insights? why not directly use the distance.**
> > >
> > >
> > > **A6:**
> > > Thank you for your valuable feedback. In our consideration, using a Gaussian filter is necessary for the following reasons.
> > >
> > > - In the process of human visual perception of the video, the processing of temporal relationships between video clips is complex, we must choose a non-linear modeling approach.
> > > -  Brain processing of temporal relationships between video clips has four characteristics: 1) Nonlinearity; 2) High-dimensional; 3) Short-term Effect; 4) Relative Temporal Distance Decay. The current video clip has $h$ Gaussian filters on both temporal directions that gradually decrease in correlation, which can reflect the visual persistence phenomenon when people browse videos. That is, the recent past has a significant impact on their perception of the present.
> > >
> > > Due to the **dynamic nature, continuity (high-dimensional), and nonlinearity (complexity)** of visual perception transmission, the discrete scalar $d_{ij}$ is insufficient to express these characteristics.
> > > Therefore, we use filter generating networks to generate filters dynamically.
> > > The Gaussian function has already been exploited in deep neural networks, such as Gaussian kernel grouping and Gaussian radial basis function. These Gaussian functions have been proven to be effective in simulating the high-dimensional nonlinear information in various scenes.
> > > Inspired by these works, we use multi-kernel Gaussian radial basis to extend the influence of $d_{ij}$ into high-dimensional space, thereby reflecting the continuous complexity of the perception process. Meanwhile, using a multi-kernel Gaussian with different biases can avoid a plateau at the beginning of training due to the highly correlated Gaussian filters.
> > >
> > >
> > >
> > > > **Q7: Is there any chance also leverage the audio signal into this work, formulating a multi-model graph?**
> > >
> > > **A7:**
> > > Thank you for your valuable suggestion. Your suggestion is very insightful and prescient. Yes, we agree that there is a chance. The original intention of our work is to solve the query-driven video content understanding, so we choose to model the video content in the graph after the global view with the ResMLP module, and the experiments achieve well results.
> > > In further research, if we interact with the original audio signal and video, we may have to perform multi-task learning with mutual guidance between modalities, such as video-guided audio graph learning, audio-guided video graph learning, and audio-video joint graph learning to promote video event understanding (localization). More utilization of semantic consistency and complementarity between modalities is needed.
> > > We will continue to study the possibility of the audio-video multi-model graph in our future work, which will be very meaningful.
> > >
> > > We hope our response addresses your concerns. If there are any further questions, please do not hesitate to let us know. Thank you for your time and valuable feedback.

---

> ### Comment · Reviewer_TT77 · 2023-11-21
> **Post rebuttal comment**
>
> Thanks all authors for their effort in the rebuttal.
>
> The rebuttal address most of my concerns, so I keep my rating suggesting acceptance of this work.
>
> Thanks

---

### Official Review · Reviewer_1gED · 2023-11-06

**Soundness:** 3 good
**Presentation:** 3 good
**Contribution:** 3 good
**Rating:** 6
**Confidence:** 4

**Summary:**

This work proposes the Unified Static and Dynamic Network (UniSDNet) for video grounding, which establishes semantic associations between multiple text/audio queries and video content in a cross-modal environment. UniSDNet combines static and dynamic modeling techniques. For static modeling, it employs an MLP within a residual structure (ResMLP) to facilitate comprehensive interactions between video content and multiple queries, enhancing mutual semantic understanding. For dynamic modeling, the paper draws inspiration from human visual perception mechanisms and constructs a diffusive connected video clip graph to represent short-term relationships and employs a multi-kernel Temporal Gaussian Filter for complex visual perception simulation. UniSDNet achieves state-of-the-art performance on various NLVG and SLVG datasets.

**Strengths:**

1. The proposed Dynamic Temporal Filter Network captures more fine-grained context correlations between video clips based on a well-desgined graph network.
2. The proposed method achieves state-of-the-art performance on NLVG and SLVG tasks.
3. In this work, two new SLVG datasets are collected based on existing NLVG datasets.
4. Compared with previous multi-queried methods, the proposed UniSDNet has less model parameters and is more efficient according to the average inference time per query.

**Weaknesses:**

1. In ResMLP, visual features and multiple query features are concatenated and fed into the network, largely leveraging the information leakage between different queries (because the features incorporate more accurate textual information that describes the video content). If each query is individually input into the network, would this method exhibit a significant performance degradation?
2. In the ablation study, individually employing the static network and DTFNet yields significant improvements compared to the baseline. However, the combination of both modules does not exhibit a notably large improvement compared to using either single module. Is there a specific explanation for this phenomenon? The authors should provide more details about the baseline models.

**Questions:**

I have listed my major concerns and questions in the weaknesses. I hope the authors can provide more details of baseline models in the ablation study and some experimental results about comparing multi-query input and single-query input.

---

> ### Author Response · Authors · 2023-11-21
> **The Response to Reviewer 1gED (Part 1/2)**
>
> Thanks for your recognition of our proposed Dynamic Temporal Filter Network, and your insightful reply is very helpful. We follow your advice and add the new experimental results about comparing multi-query input and single-query input.
>
> > **Q1: In ResMLP, visual features and multiple query features are concatenated and fed into the network, largely leveraging the information leakage between different queries (because the features incorporate more accurate textual information that describes the video content). If each query is individually input into the network, would this method exhibit a significant performance degradation?**
>
> **A1:** Thank you for your valuable feedback. Here, we attempt to clarify our conclusion from the following two aspects. On one hand, regardless multi-query or single-query modes, the role of Multilayer perceptron with residual design (ResMLP) is designed to capture the associations between query and video, simulating the brain's processing of multi-source information in the early stage of visual perception for the video. On the other hand, yes, we agree with your opinion that multiple queries can indeed provide more semantics to the video and significantly fill the gap between vision-language modalities, aiding in understanding video content.
> To eliminate your concern, we have supplemented the relevant experiments and found two facts in **Table R1**:
>
> - Our model still performs best in single-query input mode, compared to other single query methods. For example, our R@1, IoU@0.7 is 32.25, exceeding the current SOTA methods.
>
> - At present, there are some multi-query methods, but their advantages over the single-query methods are not obvious. How to better utilize multiple queries to assist in video grounding is also a challenge. Our method has advantages in handling both single-query and multi-query input modes.
>
> **Table R1: Model size vs. Infer. Speed vs. Performance comparison in different query input modes.**
>
> |(#Query) | Method | Model Size | Infer. Speed (s/Query) | R@1, IoU@0.3 | R@1, IoU@0.5 | R@1, IoU@0.7  |  R@5, IoU@0.3 | R@5, IoU@0.5  | R@5, IoU@0.7 | mIoU  |
> | :---------- | :---------- | :---------- | :----------: | :----------: | :----------: | :----------: | :----------: | :----------: | :----------: | :----------:|
> | Single | 2D-TAN  |  21.62M  |  0.061 | 59.45 | 44.51 | 26.54 | 85.53 | 77.13 | 61.96 | - |
> | Single | MS-2D-TAN  | 479.46M  |  0.141 | 61.04 | 46.16 | 29.21 | 87.30 | 78.80 | 60.85 | - |
> | Single | MSAT       |  37.19M  |  0.042 |   -   | 48.02 | 31.78 |   -   | 78.02 | 63.18 | - |
> | **Single** |**UNiSDNet**|**76.52M**|**0.009** |**68.66**|**52.35**|**32.25**|**89.74**|**83.35**|**70.61**|**50.22**|
> | Multi  | MMN       | 152.22M | 0.014 | 65.05 | 48.59 | 29.26 | 87.25 | 79.50 | 64.76 | - |
> | Multi  | PTRM      | 152.25M | 0.038 | 66.41 | 50.44 | 31.18 |   -   |   -   |   -   | 47.68 |
> | **Multi**  | **UNiSDNet**  |**76.52M**|**0.009**|**75.85**|**60.75**|**38.88**|**91.16**|**85.34**|**74.01**|**55.47**|

---

> ### Author Response · Authors · 2023-11-21
> **The Response to Reviewer 1gED (Part 2/2)**
>
> > **Q2: In the ablation study, individually employing the static network and DTFNet yields significant improvements compared to the baseline. However, the combination of both modules does not exhibit a notably large improvement compared to using either single module. Is there a specific explanation for this phenomenon? The authors should provide more details about the baseline models.**
>
> **A2:** Thank you for raising the concerns about the main network modules.
> In our manuscript, we propose a method that starts with static global review queries and video content (ResMLP), and then dynamically filters video content to extract important information (DTFNet).
> On the one hand, observing the experimental results of our static module alone (ResMLP) compared with existing works (in Tables 1 and 4 of our manuscript), previous works have overlooked and performed insufficiently in extracting information from global static reviews.
> On the other hand, even if the static review information is missed and only basic cross-modal information is obtained, we achieve good video grounding results using the dynamic module (DTFNet) alone. **Effective information filtering method (DTFNet)** can still extract useful information.
>
> For convenience, we restate Table 4 of our manuscript in the following **Table R2**. Relatively speaking, the large performance improvement of a single module through the respective exploration of multimodal information has reached **saturation** (for NLVG task, the R@1, IoU@0.3 of ResMLP and DTFNet are 73.57 and 74.56, respectively.).
> In this case, the R@1, IoU@0.3 of a single module increases from 73.57/74.56 to the final static-dynamic combination of 75.85, which is not a small improvement.
> That's why the effect of combining ResMLP and DTFNet is not as much improved as employing them alone.
> This further demonstrates the effectiveness of our static and dynamic modules.
>
> **Table R2: Ablation studies on the static and dynamic modules on the ActivityNet Captions and ActivityNet Speech datasets for both NLVG and SLVG.**
> |Task | Static Module | Dynamic Module | R@1, IoU@0.3 | R@1, IoU@0.5 | R@1, IoU@0.7  |  R@5, IoU@0.3 | R@5, IoU@0.5  | R@5, IoU@0.7 | mIoU  |
> | :----------: | :----------: | :----------: | :----------: | :----------: | :----------: | :----------: | :----------: | :----------: | :----------: |
> | NVLG | N | N | 61.22 | 44.46 | 26.76 | 87.19 | 78.63 | 63.60 | 43.98 |
> | NVLG | Y | N | 73.57 | 58.70 | 37.07 | 91.17 | 85.55 | 73.98 | 54.06 |
> | NVLG | N | Y | 74.56 | 59.45 | 37.44 | 90.98 | 85.43 | 73.60 | 54.43 |
> | **NVLG** | **Y** | **Y** | **75.85** | **60.75** | **38.88** | **91.16** | **85.34** | **74.01** | **55.47** |
> | SLVG | N | N | 53.63 | 35.91 | 20.51 | 84.71 | 74.21 | 55.95 | 38.23 |
> | SLVG | Y | N | 69.71 | 53.75 | 31.26 | 90.42 | 84.11 | 70.82 | 50.69 |
> | SLVG | N | Y | 71.34 | 54.03 | 31.51 | 89.75 | 82.62 | 68.12 | 50.97 |
> |**SLVG** | **Y** | **Y** | **72.27** | **56.29** | **33.29** | **90.41** | **84.28** | **72.42** | **52.22** |
>
>
> **About the baseline model**, we belong to the scope of the 2D proposal-based method. Specifically, we adopt the same backend decoding method as MMN (including the 2D proposal generation module). For training losses, we use the cross entropy loss between the predicted 2D proposal score map and groundtruth which is consistent with 2D-TAN, and add the contrastive loss proposed in MMN as an auxiliary constraint. That is to say, our method is based on the typical 2D temporal map's backend decoding architecture (2D-TAN and MMN), and by incorporating a static-dynamic module design as the fore-end implementation.
> Our work further develops the 2D proposal-based method in this field.
>
> We hope our response addresses your concerns. If there are any further questions, please let us know. Thank you for your time and valuable feedback.

---

> > ### Comment · Reviewer_1gED · 2023-11-23
> >
> > Thanks all authors for their effort in rebuttal. I think that Most of my concerns have been addressed. I would keep my rating and recommend the acceptance of this work.

---

### Author Response · Authors · 2023-11-21
**Global Response**

To Dear PC, AC, and All Reviewers,

First and foremost, we sincerely appreciate your time and efforts during our submission process,  allowing us to witness such a high standard and efficient peer review.
We are delighted that all reviewers recognized the merits of this paper, including novelty (Reviewers '1gED', 'TT77', 'Z5of', 'GE2s'), clarity (Reviewers 'TT77', 'GE2s'), comprehensive experiments and good performance (Reviewers '1gED', 'TT77', 'Z5of', 'GE2s').
Your positive recognition of our method UniSDNet's good universality and efficiency on both NLVG and SLVG tasks, and the contributions of two new datasets are very encouraging and deeply appreciated.



We sincerely thank all reviwers' constructive comments and suggestions. We have carefully considered all the concerns expressed by the reviewers and conducted necessary experimental verification. They are exceedingly helpful for us to improve our paper. We will continue to conduct in-depth research, explore more valuable discoveries, and do our best to contribute to the research community.
In the following, your comments are first stated and then followed by our point-by-point responses.

---

### Meta-Review · Area_Chair_kkAa · 2023-12-11

**Metareview:**

The reviewers acknowledged the strengths of the paper as follows: 1) the extension from NLVG to SLVG with a newly proposed benchmark, 2) strong empirical performance of the proposed approach on both NLVG and SLVG tasks. However, they also expressed concerns about: 1) a few writing issues leading to unclear or confusing claims, and 2) marginal improvement by the proposed Dynamic Filter Graph module.

The rebuttal addressed most of the initial concerns and questions, but the authors' explanation of how the human visual system inspired their approach remains unclear. This meta-reviewer found the argument by reviewer 'Z5of' convincing and agrees that the bio-connection could lead to confusion for some readers.

Regarding concern #2 (marginal improvement), the authors provided additional results (Table R5). However, it reveals that DTFNet provides marginal improvement over the much simpler MLP, weakening the evidence for the claimed technical contribution.

After the rebuttal, three of the reviewers remained lukewarm about the paper. Considering the lingering concerns and the high standards of ICLR, we recommend rejection at this time. However, we strongly encourage the authors to incorporate the feedback from this round of review and continue improving the manuscript for a future submission.

**Justification For Why Not Higher Score:**

The writing could be improved to make the paper much clearer. Also, the importance of DTFNet should be clearly justified.

**Justification For Why Not Lower Score:**

N/A

---

### Decision · Program_Chairs · 2024-01-16

Reject